# A Function-Centric Graph Neural Network Approach for Predicting Electron Densities

**Manuel V. Klockow, Marc K. Ickler, Peter Lippmann & Fred A. Hamprecht**
Interdisciplinary Center for Scientific Computing (IWR), Heidelberg University
69120 Heidelberg, Germany
`{manuel.klockow, marc.ickler, peter.lippmann, fred.hamprecht}`
`@iwr.uni-heidelberg.de`

## Abstract

Electronic structure predictions are relevant for a wide range of applications, from drug discovery to materials science. Since the cost of purely quantum mechanical methods can be prohibitive, machine learning surrogates are used to predict the results of these calculations. This work introduces the Basis Overlap Architecture (BOA), an equivariant graph neural network architecture based on a novel message passing scheme that utilizes the overlap matrix of the basis functions used to represent the predicted ground state electron density. BOA is evaluated on QM9 and MD density datasets, surpassing the previous state of the art in predicting accurate electron densities. Excellent generalization to larger molecules of up to nearly 200 atoms is demonstrated using a model trained only on QM9 molecules of at most 9 heavy atoms.

## 1 Introduction

Accurate electronic structure predictions are crucial for the development of new catalysts, improved batteries, or more specific drugs. Today's gold standard for reasonably sized systems is Kohn-Sham density functional theory (KS-DFT). It accounts for a significant fraction of worldwide supercomputing time, and three of its cornerstone methods are amongst the ten most cited publications of all time and fields (Van Noorden, 2025). Still, its computational cost prohibits routine use on large systems, or in very high throughput scenarios. In response, machine learning surrogates are developed to either circumvent or speed up KS-DFT calculations.

These methods range from property prediction (directly predicting observables from molecular geometry (Batatia et al., 2022; Ko et al., 2023; Kozinsky et al., 2023; Simeon & Fabritiis, 2023; Batatia et al., 2025; Liao et al., 2023; Anstine et al., 2025; Wood et al., 2025)) to incorporating more physical knowledge into the model. Examples of the latter include Hamiltonian prediction (Zhang et al., 2024a; Yuan et al., 2024; Wang et al., 2024; Qian et al., 2025; Luo et al., 2025), mimicking the self-consistent iterations of KS-DFT (Song & Feng, 2024), or predicting the electron density using orbital-free schemes (Remme et al., 2023; Zhang et al., 2024b; Remme et al., 2025). Directly predicting the electron density, as done here, lies between these extremes. The ground state electron density is an observable of central interest as it uniquely determines all ground state molecular properties in theory (Hohenberg & Kohn, 1964), and many properties of interest can be derived from the density in practice. Another use for electron densities is to reduce the number of self-consistent field iterations required in KS-DFT (Koker et al., 2024; Sunshine et al., 2023; Elsborg et al., 2025). Even a single KS-DFT diagonalization step can be enough to reach chemically accurate energies relative to a self-consistent KS solution (Jørgensen & Bhowmik, 2022; Li et al., 2025).

Most previous work on directly predicting ground state electron densities can be broadly grouped into two categories based on the representation used: The first class is based on a representation of the density in a (typically atom-centered) basis (Cuevas-Zuviría & Pacios, 2021; Rackers et al., 2023; del Rio et al., 2023; Elsborg et al., 2025; Qiao et al., 2022; 2020; Fu et al., 2024; Cheng & Peng, 2023; Kim & Ahn, 2024; Mitnikov & Jacobson, 2024; Febrer et al., 2025). A model would then predict the coefficients of an expansion of the ground state electron density in the given basis. This approach has the advantage of being relatively scalable, as the density only needs to be evaluated

on a volumetric grid at the end of the model, if at all. The choice of basis is however crucial, and to achieve high accuracy a very large number of basis functions is typically required. This problem was partially addressed by Fu et al. (2024), where virtual nodes are employed, and the basis functions adapted in a fine-tuning step to mitigate the impact of a suboptimal basis. Virtual nodes are placed in the molecule, e.g. at the midpoint of bonds, and additional localized basis functions are centered at these nodes. Another approach to address this problem was taken by Elsborg et al. (2025), where floating basis functions are employed. These floating basis functions are no longer centered at the atoms of the molecule and instead the basis function positions are predicted per molecule individually, enabling a much more flexible representation of the density.

The second class of methods works directly with a representation of the electron density on a volumetric grid (Jørgensen & Bhowmik, 2022; Li et al., 2025; Koker et al., 2024; Gong et al., 2019; Sunshine et al., 2023). While this approach avoids basis set related inaccuracies, it is typically much more memory intensive, due to the large number of grid points required to accurately represent the density.

**Contributions**  This work falls into the first category of methods, i.e. we represent the density in an atom-centered basis and the model predicts coefficients in this basis. However, in contrast to previous work, we choose to represent the density in a quadratic expansion of the basis functions, inspired by the internal representation of the density in KS-DFT calculations using a density matrix. A product of two atom-centered Gaussian-type basis functions, as employed in this work, will be centered between the two atoms, avoiding the need for floating orbitals or virtual nodes. For a benzene molecule, this is shown in Figure 1C. This work avoids explicitly predicting coefficients for each pair of basis functions, effectively employing a low-rank representation of blocks of the density matrix, without constructing the full coefficient matrix at any point.

To effectively utilize this representation, we introduce the Basis Overlap Architecture (BOA), a novel equivariant message passing neural network. The full BOA architecture is shown in Figure 1A. The fundamental idea of BOA is to imbue the model with information about the underlying basis by *interpreting*, at appropriate points in the model, the internal features *as functions* represented in the given basis. Most notably the message passing formulation utilizes the overlap between functions represented in atom-centered basis functions at different atoms to facilitate communication between nodes. During message passing each message is transformed from the basis of the sending node to the basis of the receiving node, essentially giving the best fit of the message in the basis of the receiving node. This instills the model with information not only about the underlying basis, but also the geometry of the molecule, as the overlap between basis functions centered at different atoms depends on their relative position. This novel approach to message passing is shown to be very effective for the task of predicting the electron density.

The internal features of BOA are separated into node and edge features, with the bulk of the computation happening in the node features. This separation avoids the high computational costs of basing the full procedure solely on edge features, while still allowing for a rich representation needed to accurately predict the density. The flow of information between node and edge features is shown in Figure 1B.

BOA is evaluated on electron densities generated from the widely used QM9 dataset (Jørgensen & Bhowmik, 2022; Li et al., 2025; Ruddigkeit et al., 2012; Ramakrishnan et al., 2014) and a molecular dynamics (MD) dataset of small organic molecules (Cheng & Peng, 2023; Bogojeski et al., 2020; Brockherde et al., 2017). On all evaluated datasets BOA outperforms previous state-of-the-art methods by a significant margin. The full code to reproduce these results is available at `https://github.com/sciai-lab/boa`.

## 2 METHODS

To achieve the desired inductive bias, internal features are interpreted as functions represented in an atom-centered basis. At each point, the node features are given by $h_{am\mu}$, where $a \in \mathcal{N}$ is the node with atom type $Z_a$, $\mathcal{N}$ is the set of all nodes, $m \in \{1, \ldots, N^c\}$ is the feature channel, and $\mu \in \{1, \ldots, N_{Z_a}^B\}$ is the basis index. We use Gaussian-type ("GTO") basis functions, specifically a fully uncontracted version of `def2-QZVPPD` (Weigend & Ahlrichs, 2005). A correction to the radial part is predicted by the model; see Appendix A for details. The internal features $h_{am\mu}$ are

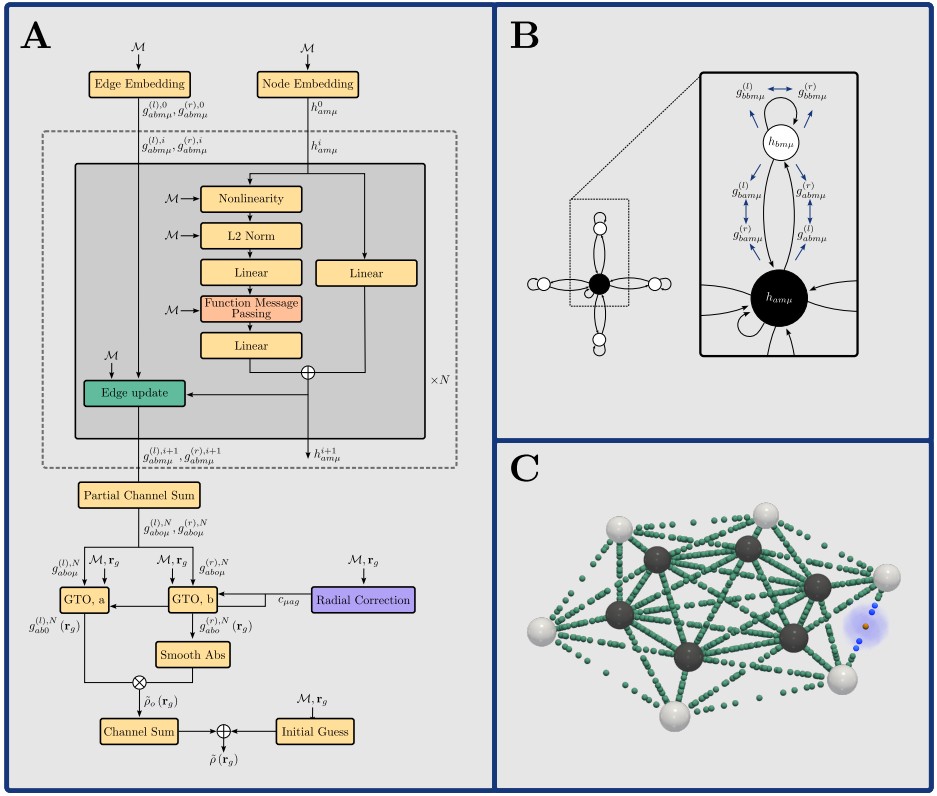

Figure 1: **The Basis Overlap Architecture (BOA). (A)** The node embeddings are updated using BOA blocks, which contain a function message passing step to facilitate communication between nodes. The edge features are modified using the edge update block, which uses the current edge and node features to calculate new edge features. The output of the BOA backbone consists of coefficients used to expand the density in atom-centered Gaussian-type basis functions (Sec. 2.1). In the partial channel mean the number of channels is reduced by taking the mean of groups of channels. $\mathcal{M}$ denotes the molecular geometry and $\mathbf{r}_g$ are the grid positions with $g \in \{1, \dots, N^g\}$ and $N^g$ being the number of grid points. **(B)** shows the flow of information in the BOA message passing block. While the edge features are updated using the node features, there is no flow of information from the edge features to the node features. Every edge in the graph has two edge features with superscript $(l)$ and $(r)$ respectively, corresponding to the two adjacent nodes. **(C)** shows a benzene molecule from the MD dataset with carbon atoms (black) and hydrogen atoms (white). Shown as green spheres are the centers of the Gaussian parts of the products of the atom-centered basis functions. The smaller `def2-SVP` (Weigend & Ahlrichs, 2005) basis set is used here for better visualization. A product of $l = 0$ basis functions from two different hydrogen atoms is shown in blue, with its center (yellow) lying in the middle between the two atoms on which the original basis functions are centered. Since the products of basis functions are well distributed in space, a highly accurate representation of the density can be achieved without the need for floating basis functions or virtual nodes.

understood as expansion coefficients of $N^c$ functions of space

$$h_m(\mathbf{r}) = \sum_{a \in \mathcal{N}} h_{am}(\mathbf{r}) = \sum_{a \in \mathcal{N}} \sum_{\mu} h_{am\mu} \omega_\mu^{Z_a}(\mathbf{r} - \mathbf{r}_a), \tag{1}$$

consisting of a sum over functions $h_{am}(\mathbf{r})$ localized at the node positions $\mathbf{r}_a$. Each of these functions in turn is represented in the atom type specific basis $\omega_\mu^{Z_a}(\mathbf{r})$. The same principle holds for the edge features $g_{abm\mu}^{(l)}$ and $g_{abm\mu}^{(r)}$, where $(a, b) \in \mathcal{E}_e$ is a directed edge in the graph, with $\mathcal{E}_e$ being the set of all edges. Superscripts $l$ and $r$ denote left and right, respectively, for reasons that are clear from Eq. 2. The $l$ edge features are interpreted as functions localized at node $a$, while the $r$ edge features are interpreted as functions localized at node $b$.

To guarantee rotational equivariance the choice of basis functions is essential. Choosing basis functions where the angular part is given by spherical harmonics ensures that the basis function coefficients transform according to irreducible representations of SO(3), making it possible to construct equivariant operations. BOA is fully equivariant under rotations and translations by construction (see App. H).

## 2.1 DENSITY REPRESENTATION

Previous work explored different ways to represent the density, either in a basis or directly on the grid. We choose a representation in an atom-centered basis. In contrast to previous work, we however do not expand the density (Cuevas-Zuviría & Pacios, 2021; Rackers et al., 2023; del Rio et al., 2023; Qiao et al., 2022; Fu et al., 2024; Cheng & Peng, 2023) or its square root (Mitnikov & Jacobson, 2024) directly as a linear expansion of the basis functions $\rho(\mathbf{r}) = \sum_{a \in \mathcal{N}} \sum_{\mu} p_{a\mu} \omega_{\mu}^{Z_a}(\mathbf{r} - \mathbf{r}_a)$. Instead, we choose a quadratic expansion, inspired by the natural expansion of the density in squared orbital functions in KS-DFT:

$$\rho(\mathbf{r}) = \sum_{a \in \mathcal{N}} \hat{g}_a^{(l)}(\mathbf{r})\hat{g}_a^{(r)}(\mathbf{r}) + \sum_{(a,b) \in \mathcal{E}_e} \sum_o^{N^o} g_{abo}^{(l)}(\mathbf{r})g_{abo}^{(r)}(\mathbf{r}), \qquad (2)$$

where $N^o$ is the number of function pairs per edge and $g_{abo}^{(l)}(\mathbf{r})$, $g_{abo}^{(r)}(\mathbf{r})$, $\hat{g}_a^{(l)}(\mathbf{r})$ and $\hat{g}_a^{(r)}(\mathbf{r})$ are expanded in the localized basis functions $\omega_{\mu}^{Z_a}(\mathbf{r})$ and $\omega_{\mu}^{Z_b}(\mathbf{r})$:

$$g_{abo}^{(l)}(\mathbf{r}) = \sum_{\mu} g_{abo\mu}^{(l)}\omega_{\mu}^{Z_a}(\mathbf{r} - \mathbf{r}_a), \quad g_{abo}^{(r)}(\mathbf{r}) = \sum_{\mu} g_{abo\mu}^{(r)}\omega_{\mu}^{Z_b}(\mathbf{r} - \mathbf{r}_b), \qquad (3)$$

$$\hat{g}_a^{(l)}(\mathbf{r}) = \sum_{\mu} \hat{g}_{a\mu}^{(l)}\omega_{\mu}^{Z_a}(\mathbf{r} - \mathbf{r}_a), \qquad \hat{g}_a^{(r)}(\mathbf{r}) = \sum_{\mu} \hat{g}_{a\mu}^{(r)}\omega_{\mu}^{Z_a}(\mathbf{r} - \mathbf{r}_a). \qquad (4)$$

This representation can be rewritten by defining $\Gamma_{ab\mu\nu} = \sum_o^{N^o} g_{abo\mu}^{(l)}g_{abo\nu}^{(r)} + \delta_{ab}\hat{g}_{a\mu}^{(l)}\hat{g}_{a\nu}^{(r)}$, with $\delta_{ab} = 1$ if $a = b$ and 0 otherwise, resulting in $\rho(\mathbf{r}) = \sum_{(a,b) \in \mathcal{E}_e} \sum_{\mu\nu} \Gamma_{ab\mu\nu}\omega_{\mu}^{Z_a}(\mathbf{r} - \mathbf{r}_a)\omega_{\nu}^{Z_b}(\mathbf{r} - \mathbf{r}_b)$. Each $\Gamma_{ab\mu\nu}$ can be interpreted as a block of the full density matrix $\Gamma_{\mu\nu}$, where the indices $\mu$ and $\nu$ are understood to run over all basis functions of all atoms. Using the full density matrix, the density is represented by $\rho(\mathbf{r}) = \sum_{\mu\nu} \Gamma_{\mu\nu}\bar{\omega}_{\mu}(\mathbf{r})\bar{\omega}_{\nu}(\mathbf{r})$, where $\bar{\omega}(\mathbf{r})$ is the concatenation of the basis functions of all atoms. This is exactly the representation of the density used in KS-DFT calculations. The full density matrix $\Gamma_{\mu\nu}$ is never explicitly constructed in BOA, instead the functions $g_{abo}^{(l)}(\mathbf{r})$ and $g_{abo}^{(r)}(\mathbf{r})$ are evaluated on the grid and the density is obtained using Eq. 2, avoiding the costly evaluation of all pairwise products of basis functions on the grid. Equation 2 amounts to evaluating a density given by a low-rank representation of each block $\Gamma_{ab\mu\nu}$ of the density matrix.

The expansion coefficients $g_{abo\mu}^{(l)}$ and $g_{abo\mu}^{(r)}$ are predicted from the molecule geometry $\mathcal{M}$ by BOA as described in the following sections. The self-loop coefficients $\hat{g}_{a\mu}^{(l)}$ and $\hat{g}_{a\mu}^{(r)}$ are used to represent an initial guess for the density, which is added to the density predicted by BOA. Taking the smooth absolute value of one of the functions in each pair proved beneficial (see App. B).

## 2.2 INITIAL GUESS OF THE ELECTRON DENSITY

BOA uses an initial guess of the ground state electron density based on the atom types of the nodes. After pre-training the initial guess, only an offset to that guess is learned by the model. The learned guess amounts to one pair of edge functions as described in Section 2.1 per atom. Consequently, the initial guess for the density is given by $\sum_{a \in \mathcal{N}} \hat{g}_a^{(l)}(\mathbf{r})\hat{g}_a^{(r)}(\mathbf{r})$ and is part of the predicted electron density as shown in Eq. 2. The initial guess coefficients are pre-trained for 1000 steps, but are not fixed after that, i.e. the initial guess is refined during the full training process.

## 2.3 NODE AND EDGE EMBEDDINGS

The node and edge embeddings are used to set the initial node and edge features respectively. Both are lookup tables, which map the atom type $Z_a$ of the node $a$ and the atom types of the nodes

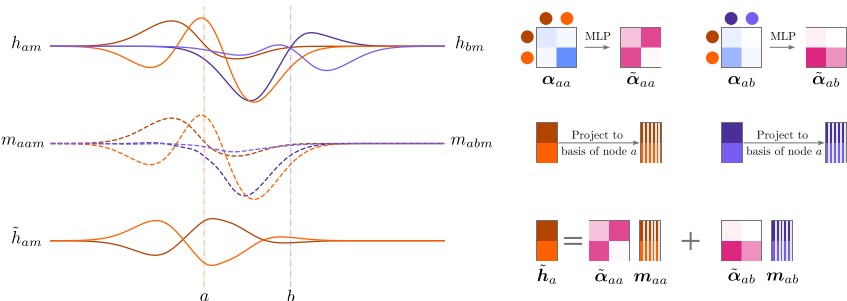

Figure 2: **1D illustration of function message passing.** Two nodes $a$ and $b$ are shown, each with two feature functions (blue and orange) centered at their respective positions. These functions are expanded in a local basis built from monomials multiplied with Gaussians. The first line shows the original feature functions $h_{am}$ and $h_{bm}$. The second line shows the messages sent from node $a$ and node $b$ to node $a$. These are the projection of the respective feature functions to the basis of node $a$. The third line shows the new features $\tilde{h}_{am}$ after message passing which are a weighted sum of all incoming messages.

connected by the edge $(a, b)$ to a learned set of features. To keep equivariance under rotations, only the coefficients corresponding to $l = 0$ basis functions are set in the embeddings, while the rest is initialized to zero. The node features are therefore initialized as

$$h_{am\mu} = \begin{cases} W^{(n)}_{Z_a m\mu}, & \text{if } \omega^{Z_a}_\mu(\mathbf{r}) \text{ is an } l_{Z_a\mu} = 0 \text{ basis function} \\ 0, & \text{otherwise,} \end{cases} \tag{5}$$

where $W^{(n)}_{Z_a m\mu}$ is learned during training and $l_{Z_a\mu}$ is labeling the irreducible representation of the basis function $\omega^{Z_a}_\mu(\mathbf{r})$. The edge features are initialized similarly as

$$g^{(l)}_{abm\mu} = \begin{cases} W^{(e,l)}_{Z_a Z_b m\mu}, & \text{if } \omega^{Z_a}_\mu(\mathbf{r}) \text{ is an } l_{Z_a\mu} = 0 \text{ basis function} \\ 0, & \text{otherwise,} \end{cases} \tag{6}$$

$$g^{(r)}_{abm\mu} = \begin{cases} W^{(e,r)}_{Z_a Z_b m\mu}, & \text{if } \omega^{Z_b}_\mu(\mathbf{r}) \text{ is an } l_{Z_b\mu} = 0 \text{ basis function} \\ 0, & \text{otherwise,} \end{cases} \tag{7}$$

where $W^{(e,l)}_{Z_a Z_b m\mu}$ and $W^{(e,r)}_{Z_a Z_b m\mu}$ are again learned.

## 2.4 BASIS OVERLAP MESSAGE PASSING

The fundamental principle behind the message passing mechanism introduced in this section is that each feature channel should be interpretable as a function represented in a given basis. To respect this interpretation, a basis change is employed to transform each message from the basis of the sending node to the basis of the receiving node, a concept similar to frame-to-frame transitions employed in local canonicalization schemes (Lippmann et al., 2025). Inspired by the attention mechanism (Vaswani et al., 2017), the messages are weighted by attention weights, which are calculated from the overlap integrals between the features of the sending and receiving node. The full message passing mechanism is shown in Figure 3B and a one dimensional illustration is given in Figure 2.

BOA operates using two different cutoff radii $r_e, r_{mp} \in \mathbb{R}$, with $r_e$ being a smaller cutoff defining the edges $\mathcal{E}_e$ used for the edge features and in the edge update module defined in 2.8, and $r_{mp}$ being a larger cutoff defining the edges $\mathcal{E}_{mp}$ used in the message passing step.

The first step in transforming the features of node $b$ into the basis of node $a$ is calculating the overlap integrals between these features and the basis functions of node $a$. This overlap $o_{abm\mu}$ is given by

$$o_{abm\mu} = \int d\mathbf{r}\, \omega^{Z_a}_\mu(\mathbf{r} - \mathbf{r}_a) h_{bm}(\mathbf{r}) = \int d\mathbf{r}\, \omega^{Z_a}_\mu(\mathbf{r} - \mathbf{r}_a) \sum_\nu \omega^{Z_b}_\nu(\mathbf{r} - \mathbf{r}_b) h_{bm\nu} \tag{8}$$

$$= \sum_\nu W^{ab}_{\mu\nu} h_{bm\nu}, \tag{9}$$

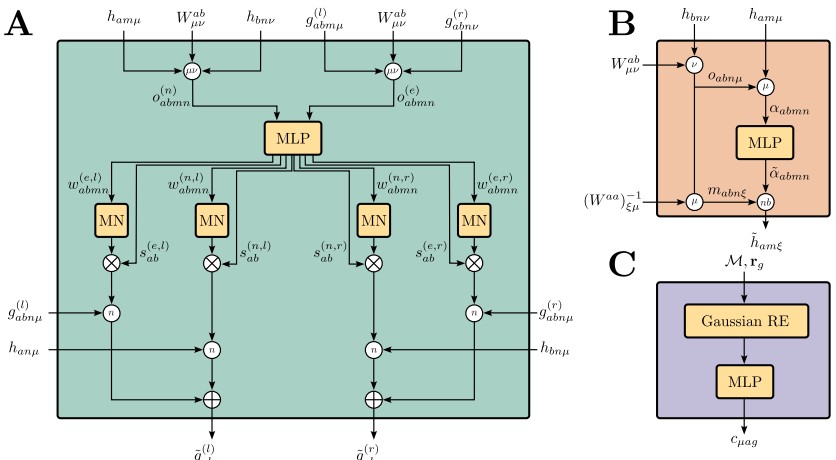

Figure 3: **Subblocks of the BOA architecture.** An encircled index in the graph indicates a contraction over the corresponding dimension, i.e. multiplication of the inputs and summation over the given index. **(A)** The new edge features $\tilde{g}^{(l)}_{abm\mu}$ and $\tilde{g}^{(r)}_{abm\mu}$ are generated as a superposition of the old edge features $g^{(l)}_{abm\mu}$ and $g^{(r)}_{abm\mu}$ and the node features $h_{am\mu}$ and $h_{bm\mu}$. The weights of this superposition are generated by passing the overlap integrals of the edge and node features through an MLP. After the MLP the weights are normalized using the matrix normalization (MN) block and scaled by the predicted factors $s^{(\cdot,\cdot)}_{ab}$. The resulting weight matrices are then used to linearly mix the edge and node features. **(B)** Incoming messages are calculated by transforming the feature functions from the basis of the sending node to the basis of the receiving node. Attention weights $\tilde{\alpha}_{abmn}$ are calculated from the overlap between the feature functions of the two nodes and used to weight the messages. **(C)** A correction to the radial part of the basis functions is learned for each atom type separately by a small MLP. The radius is passed through a Gaussian radial embedding (Passaro & Zitnick, 2023) before being fed into the MLP. The output $c_\mu$ of the MLP is then used to scale the radial part of the basis functions by $1 + c_\mu$ (see App. A for details).

where $W^{ab}_{\mu\nu} = \int d\mathbf{r}\, \omega^{Z_a}_\mu(\mathbf{r} - \mathbf{r}_a)\omega^{Z_b}_\nu(\mathbf{r} - \mathbf{r}_b)$ is the overlap matrix between the basis functions of node $a$ and node $b$. From these overlap integrals, the representation of the features of node $b$ in the basis of node $a$ can be obtained by multiplying with the inverse of the overlap matrix $(W^{aa})^{-1}_{\mu\nu}$ of the basis functions of node $a$, resulting in the message $m_{abm\mu}$ from node $b$ to node $a$

$$m_{abm\mu} = \sum_\nu (W^{aa})^{-1}_{\mu\nu}\, o_{abm\nu}. \tag{10}$$

These messages $m_{abm\mu}$ are the coefficients in the basis of node $a$ that best represent the features of node $b$ in the least squares sense. A derivation of this result can be found in Appendix G.

To weight the messages, an attention matrix $\tilde{\alpha}_{abmn}$ is calculated for each edge $(a, b) \in \mathcal{E}_{\mathrm{mp}}$. Each entry in this attention matrix describes the weight of the message from channel $n$ of node $b$ to channel $m$ of node $a$. The attention matrix is calculated from the overlap between the feature functions of the two nodes

$$\alpha_{abmn} = \int d\mathbf{r}\, h_{am}(\mathbf{r})h_{bn}(\mathbf{r}) = \int d\mathbf{r} \sum_{\mu\nu} h_{am\mu}\omega^{Z_a}_\mu(\mathbf{r} - \mathbf{r}_a)h_{bn\nu}\omega^{Z_b}_\nu(\mathbf{r} - \mathbf{r}_b) \tag{11}$$

$$= \sum_{\mu\nu} h_{am\mu}W^{ab}_{\mu\nu}h_{bn\nu} = \sum_\mu h_{am\mu}o_{abn\mu}. \tag{12}$$

This feature overlap matrix $\alpha_{abmn}$ is processed by a multi-layer perceptron to produce the attention weights $\tilde{\alpha}_{abmn}$, which determine the contribution of messages from neighboring nodes. These

weighted messages are then aggregated to update the features of node $a$

$$\tilde{h}_{am\mu} = \sum_{b \text{ s.t. } (a,b)\in\mathcal{E}_{\text{mp}}} \sum_{n} \tilde{\alpha}_{abmn} m_{abn\mu}. \tag{13}$$

Since the edges $\mathcal{E}_{\text{mp}}$ include self-loops, the original features of each node are included in the message passing step, which amounts to a residual connection weighed by the attention weights $\tilde{\alpha}_{aamn}$.

## 2.5 NONLINEARITY

Searching for a suitable nonlinearity for features that should be interpretable as functions, one faces similar problems as for nonlinearities applied in equivariant neural networks. Here features are grouped together to form tensors with known transformation behavior under rotations, so that the nonlinearity can not be applied independently to each feature. In our case, the features are not only grouped to form tensors, but also form larger groups that each represent a function.

One solution in the case of equivariant networks are gated nonlinearities (Weiler et al., 2018), where first scalar features are computed to which some nonlinear function is applied. The resulting scalar features are then used to scale the tensors, i.e. each group of features forming a tensor is scaled with the same scalar. In a similar approach we first calculate scalar features

$$l_{amn} = \int d\mathbf{r} \int d\mathbf{r}' \, \frac{h_{am}(\mathbf{r})h_{an}(\mathbf{r}')}{\|\mathbf{r} - \mathbf{r}'\|} = \sum_{\mu\nu} h_{am\mu} C_{\mu\nu}^{aa} h_{an\nu}, \tag{14}$$

where $C_{\mu\nu}^{aa} = \int d\mathbf{r} \int d\mathbf{r}' \left(\omega_{\mu}^{Z_a}(\mathbf{r} - \mathbf{r}_a)\omega_{\nu}^{Z_a}(\mathbf{r}' - \mathbf{r}_a)\right) / (\|\mathbf{r} - \mathbf{r}'\|)$ is the Coulomb matrix, which can be generated for Gaussian-type basis functions using the PySCF package (Sun et al., 2020). The resulting scalar features $l_{amn}$ are flattened and passed through an MLP. The result

$$\mathbf{w}_a = \text{MLP}_{Z_a}(\mathbf{l}_a) \tag{15}$$

is reshaped to the original shape of $l_{amn}$ and then used to linearly transform the features of node $a$

$$\tilde{h}_{am\mu} = \sum_{n} w_{amn} h_{an\mu}. \tag{16}$$

$Z_a$ denotes the atom type of node $a$, i.e. separate MLPs are learned for each atom type.

## 2.6 L2 NORMALIZATION

The normalization also aims to respect the function nature of the features, so the L2 norm of each of the per-atom channel functions is used for normalization. The norm is calculated as $n_{am} = \sqrt{\int d\mathbf{r} \, (h_{am}(\mathbf{r}))^2} = \sqrt{\sum_{\mu\nu} h_{am\mu} W_{\mu\nu}^{aa} h_{am\nu}}$ and the features are normalized by $\tilde{h}_{am\mu} = h_{am\mu} / (n_{am} + \epsilon)$ where $\epsilon$ is chosen as $10^{-6}$ to avoid numerical issues.

## 2.7 LINEAR LAYERS

Standard equivariant linear layers, as implemented in the e3nn package (Geiger & Smidt, 2022), are applied to the features of each channel and node. This means that only tensors of the same type are mixed and biases are only applied to scalars to preserve equivariance. The linear layers used depend on the atom type of each node, i.e. separate parameters are learned for each atom type. After the application of these equivariant linear layers, the channels are mixed by a weight matrix, which again depends on the atom type of the node. This results in

$$\tilde{h}_{am\mu}^{Z_a} = \sum_{n} \sum_{\nu} W_{mn}^{Z_a} W_{\mu\nu}^{Z_a,(\text{eq})} h_{an\nu} \tag{17}$$

where $W_{mn}^{Z_a}$ is the weight matrix for the mixing of the channels and $W_{\mu\nu}^{Z_a,(\text{eq})}$ is the weight matrix of the equivariant linear layer. While the linear layer used here is fully equivariant, it arguably does not fully respect the function interpretation of the features. Initial experiments with linear layers that respect the function interpretation, i.e. by applying an integration $\int d\mathbf{r} \, w_{mn}(\mathbf{r})h_n(\mathbf{r})$ where $w_{mn}(\mathbf{r})$ are learned functions represented in some basis, showed strong instabilities in training. Since the linear layers shown here worked well in practice, we decided to use them in the final architecture, opting for stability over additional inductive bias.

## 2.8 EDGE UPDATE

The edge functions $g_{abm}^{(l)}(\mathbf{r}), g_{abm}^{(r)}(\mathbf{r})$ are updated after each BOA block. The flow of information is illustrated in Figure 1B. The node features $h_{am}(\mathbf{r})$ are used together with the old edge features $g_{abm}^{(l)}(\mathbf{r}), g_{abm}^{(r)}(\mathbf{r})$ to generate new edge features $\tilde{g}_{abm}^{(l)}(\mathbf{r}), \tilde{g}_{abm}^{(r)}(\mathbf{r})$. There is however no flow of information from the edge features to the node features. The full edge update procedure is shown in Figure 3A. Intermediate features are generated using the SO(3)-invariant overlap integrals of edge and node features

$$o_{abmn}^{(n)} = \int d\mathbf{r} \, h_{am}(\mathbf{r})h_{bn}(\mathbf{r}) = \sum_{\mu\nu} h_{am\mu}W_{\mu\nu}^{ab}h_{bn\nu}, \tag{18}$$

$$o_{abmn}^{(e)} = \int d\mathbf{r} \, g_{abm}^{(l)}(\mathbf{r})g_{abn}^{(r)}(\mathbf{r}) = \sum_{\mu\nu} g_{abm\mu}^{(l)}W_{\mu\nu}^{ab}g_{abn\nu}^{(r)}, \tag{19}$$

These intermediate features are flattened and passed through an MLP to obtain weights $w_{abmn}^{(n,l)}$, $w_{abmn}^{(n,r)}, w_{abmn}^{(e,l)}, w_{abmn}^{(e,r)}, s_{ab}^{(n,l)}, s_{ab}^{(n,r)}, s_{ab}^{(e,l)}, s_{ab}^{(e,r)}$. The weight matrices $\mathbf{w}_{ab}^{(n,l)}, \mathbf{w}_{ab}^{(n,r)}, \mathbf{w}_{ab}^{(e,l)}, \mathbf{w}_{ab}^{(e,r)}$ are normalized by

$$\tilde{\mathbf{w}}_{ab}^{(\cdot,\cdot)} = \left(\mathbf{w}_{ab}^{(\cdot,\cdot)}\right) \Big/ \left(\left\|\mathbf{w}_{ab}^{(\cdot,\cdot)}\right\|_f + \epsilon\right)\gamma^{(\cdot,\cdot)}, \tag{20}$$

where $(\cdot,\cdot)$ denotes either $(n,l)$, $(n,r)$, $(e,l)$ or $(e,r)$ and $\gamma^{(\cdot,\cdot)}$ is a learned scalar factor. The normalization is computed in the Frobenius norm $\|\cdot\|_f$. Using these weights the new edge features

$$\tilde{g}_{abm\mu}^{(\cdot)} = \sum_n s_{ab}^{(e,\cdot)}\tilde{w}_{abmn}^{(e,\cdot)}g_{abn\mu}^{(\cdot)} + \sum_n s_{ab}^{(n,\cdot)}\tilde{w}_{abmn}^{(n,\cdot)}h_{\star n\mu} \tag{21}$$

are generated, where $\star = a$ for the $l$ features and $\star = b$ for the $r$ features.

## 3 EXPERIMENTS

The performance of BOA is evaluated on two electron density datasets based on QM9 (Jørgensen & Bhowmik, 2022; Li et al., 2025; Ruddigkeit et al., 2012; Ramakrishnan et al., 2014) and on a dataset based on MD trajectories (Cheng & Peng, 2023; Bogojeski et al., 2020; Brockherde et al., 2017). These datasets provide ground state electron densities for given geometries on a volumetric grid. Two versions of BOA are evaluated, a small version and a large version, differing only in the number of grid points used to evaluate the loss during training, and the batch size. The small version uses 5000 grid points and a batch size of 12 while the large version uses 6000 grid points and a batch size of 24. Using these settings, the small version fits on a single A100 GPU with ∼40GB of memory while the large version requires an H100 GPU with ∼94GB of memory. All hyperparameters and training details are listed in Appendix I.

Additionally, the generalization capabilities of BOA are evaluated on larger molecules with up to almost 200 atoms from the QMugs dataset (Isert et al., 2022) not seen during training. Ground-truth ground state electron densities are generated using the same method as used for the QM9 PySCF dataset (Li et al., 2025), the dataset on which the evaluated models were trained.

### 3.1 EVALUATION ON SMALL MOLECULES

As in previous works, the predicted electron density $\tilde{\rho}(\mathbf{r})$ is compared to the reference electron density $\rho(\mathbf{r})$ using the normalized mean absolute error $\text{NMAE}(\tilde{\rho}, \rho) = \left(\int d\mathbf{r} \, |\tilde{\rho}(\mathbf{r}) - \rho(\mathbf{r})|\right) / \left(\int d\mathbf{r} \, |\rho(\mathbf{r})|\right)$. The integration is approximated on the full regular grid by summing over all grid points.

The two QM9 based datasets differ in the way the reference electron density was calculated, either using VASP (Jørgensen & Bhowmik, 2022) or PySCF (Li et al., 2025). BOA surpasses the previous state of the art on both QM9 datasets, as shown in Table 1. The split between training, validation and test set is taken from Fu et al. (2024) for the QM9 VASP dataset and from Li et al. (2025) for the QM9 PySCF dataset.

Table 1: **Comparison of BOA with previous best methods on the QM9 charge density datasets.** Two datasets based on QM9 are evaluated, differing in the reference electron density calculation method. Errors are reported as NMAE [%]. For BOA the mean and standard error over three runs are reported for the small models. For the large models the mean and standard error over five runs are reported. Errors of eqDeepDFT, InfGCN, ChargE3Net, and SCDP are reproduced from Fu et al. (2024). The ResNet results are taken from Li et al. (2025).

| NMAE [%] | VASP ground truth | PySCF ground truth |
|---|---|---|
| eqDeepDFT (Jørgensen & Bhowmik, 2022) | 0.284 | n/a |
| InfGCN (Cheng & Peng, 2023) | 0.869 | n/a |
| ChargE3Net (Koker et al., 2024) | 0.196 | n/a |
| SCDP (Fu et al., 2024) | 0.178 | n/a |
| ELECTRA (Elsborg et al., 2025) | 0.177 | n/a |
| ResNet (Li et al., 2025) | n/a | 0.14 |
| BOA small | $0.1381 \pm 0.0003$ | $0.13 \pm 0.01$ |
| BOA large | $\mathbf{0.1339 \pm 0.0005}$ | $\mathbf{0.116 \pm 0.006}$ |

Additionally, the performance of BOA is evaluated on the MD dataset. As described in Cheng & Peng (2023), the MD dataset is curated from two sources. The ethanol, benzene, phenol and resorcinol data is taken from Bogojeski et al. (2020), while the ethane and malonaldehyde data is taken from Brockherde et al. (2017). BOA outperforms all previous methods on all molecules but one, where it matches the state of the art, as shown in Table 2. Especially promising is the fact that no additional hyperparameter tuning was needed to achieve these results. The model is exactly the same as in the QM9 experiments, with the only change being a reduction in training steps to 200,000. Since only a training and test set is provided for the MD dataset, 10% of the training set is randomly sampled and used as a validation set to choose the best model during training.

Table 2: **Comparison of BOA with other methods on the MD charge density dataset.** Errors are reported as NMAE [%]. For BOA the mean and standard error of the mean over three runs are reported. Errors of the other models (InfGCN (Cheng & Peng, 2023), GPWNO (Kim & Ahn, 2024), SCDP (Fu et al., 2024), ELECTRA (Elsborg et al., 2025)) are reproduced from Elsborg et al. (2025).

| NMAE [%] | ethanol | benzene | phenol | resorcinol | ethane | malonaldehyde |
|---|---|---|---|---|---|---|
| InfGCN | 8.43 | 5.11 | 5.51 | 5.95 | 7.01 | 10.34 |
| GPWNO | 4.00 | 2.45 | 2.68 | 2.73 | 3.67 | 5.32 |
| SCDP | 2.34 | 1.13 | 1.29 | 1.35 | 2.05 | 2.71 |
| ELECTRA | 1.02 | 0.45 | **0.56** | 0.62 | 0.91 | 0.80 |
| BOA small | $\mathbf{0.710 \pm 0.004}$ | $\mathbf{0.361 \pm 0.003}$ | $\mathbf{0.56 \pm 0.03}$ | $\mathbf{0.371 \pm 0.004}$ | $\mathbf{0.772 \pm 0.002}$ | $\mathbf{0.61 \pm 0.01}$ |

### 3.2 GENERALIZATION TO LARGER MOLECULES

To evaluate the generalization performance of BOA to larger molecules, the QMugs dataset (Isert et al., 2022) is used. Ground-truth ground state electron densities for molecules with up to almost 200 atoms are calculated using PySCF (Sun et al., 2020) with the same settings as used for the QM9 PySCF dataset (Li et al., 2025). Two BOA models trained on only the QM9 PySCF dataset are evaluated on this new test set. One of the models uses a smaller message passing cutoff of $r_{\mathrm{mp}} = 3\text{Å}$ instead of $r_{\mathrm{mp}} = 6\text{Å}$ and a smaller edge feature cutoff of $r_{\mathrm{e}} = 2\text{Å}$ instead of $r_{\mathrm{e}} = 3\text{Å}$. Additionally, we evaluate the ResNet model (Li et al., 2025) on this dataset for comparison. The time-scaling behavior and the NMAE of all three models are shown in Figure 4. While the BOA model with the standard cutoffs performs worse than the ResNet model, the NMAE of the BOA model with smaller cutoffs stays roughly constant over the evaluated molecule sizes and outperforms the ResNet model. This shows that to achieve good generalization from small to large molecules, it is beneficial to limit the field of view of the model. This may be because the contents of a large field of view differ significantly between small and large molecules, introducing a significant distribution shift, making

it difficult for the model to generalize. Using a smaller field of view enables BOA to generalize well to large molecules despite being trained on only small molecules of up to 9 heavy atoms.

The time scaling behavior shown in Figure 4A also shows a second advantage of using smaller cutoffs. The BOA model with smaller cutoffs is significantly faster than both the ResNet model and the BOA model with larger cutoffs when evaluating larger molecules.

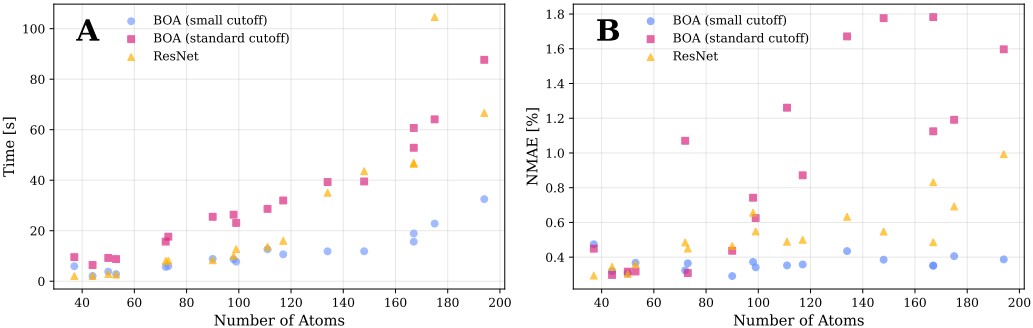

Figure 4: **Generalization of BOA to larger molecules.** Two BOA models trained on the QM9 PySCF data are evaluated on larger molecules from the QMugs dataset (Isert et al., 2022). One of the models uses a smaller message passing cutoff of $r_{\mathrm{mp}} = 3\text{Å}$ instead of $r_{\mathrm{mp}} = 6\text{Å}$ and a smaller edge feature cutoff of $r_{\mathrm{e}} = 2\text{Å}$ instead of $r_{\mathrm{e}} = 3\text{Å}$. **(A)** shows the time scaling behavior of both models compared to the ResNet model (Li et al., 2025). **(B)** shows the NMAE $[\%]$ of both models on the QMugs dataset over the number of atoms in the molecules. While the BOA model with the larger cutoffs performs worse than the ResNet model, the BOA model with smaller cutoffs outperforms the ResNet model with accuracy staying roughly constant over the evaluated molecule sizes.

## 4    DISCUSSION

Electronic structure calculations play a fundamental role in computational chemistry, with a myriad of practical applications. Accelerating these calculations has the potential of enabling new applications which are currently not possible, and of cutting costs for what is already feasible. BOA takes a step towards this goal by introducing a novel architecture, permitting predictions of the electron density with unprecedented accuracy. These results are enabled by a novel message passing mechanism treating the internal features as functions additional to a representation of the density in a quadratic basis expansion. The excellent generalization capabilities of BOA to larger molecules enables its application to problems where full electronic structure calculations are especially costly.

While BOA already achieves state-of-the-art accuracy on the evaluated datasets, there are several avenues for future work. Both the QM9 and the MD dataset contain only relatively small organic molecules. Training on larger more diverse datasets will be needed to enable applicability to a wider range of practical applications. Generalization over a large part of the periodic table would be desirable, which might require changes to the architecture. Currently, BOA uses separate parameters for each atom type, which could become infeasible for more diverse datasets. A unified basis set for all atom types could be employed in the future to mitigate this problem.

An efficient representation of the density is crucial for basis-set-based models like BOA. Another avenue for future work is therefore to improve the basis set used. BOA currently uses a fixed basis set of uncontracted Gaussian-type basis functions as a base and learns a radial correction factor. To achieve more flexibility, also the exponents of the Gaussian basis functions could be learned, as done in (Fu et al., 2024). In BOA, the overlap and Coulomb matrices used internally could be adapted during training, which could be achieved using differentiable quantum chemistry packages like PySCFAD (Zhang & Chan, 2022).

## ACKNOWLEDGMENTS

This study has received funding from the Klaus Tschira Stiftung gGmbH (Simplaix project) and the Wildcard program from the Carl Zeiss Stiftung. Also this work is supported by Deutsche Forschungsgemeinschaft (DFG) under Germany's Excellence Strategy EXC-2181/1 - 390900948 (the Heidelberg STRUCTURES Excellence Cluster). The authors acknowledge support by the state of Baden-Württemberg through bwHPC and the German Research Foundation (DFG) through grant INST 35/1597-1 FUGG.

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

## A  RADIAL CORRECTION FACTOR

While internally all features are represented in a basis consisting of Gaussian-type basis functions, for the final density representation an additional correction factor to the radial part of the basis functions is learned. For each atom type $Z$ separately a small MLP is used to predict a correction factor $c_\mu$ based on the radius $r = \|\mathbf{r}\|$. As shown in Figure 3C the radius is passed through a Gaussian radial embedding (Passaro & Zitnick, 2023) before being fed into the MLP. The corrected basis functions $\tilde{\omega}_\mu^Z(\mathbf{r})$ are then given by $\tilde{\omega}_\mu^Z(\mathbf{r}) = \omega_\mu^Z(\mathbf{r})(1 + c_\mu(r))$, where the factor $c_\mu$ is predicted per tensor, i.e. is the same for each basis function that corresponds to the same tensor to keep equivariance. The impact of this radial correction factor is evaluated in Appendix C.1.

## B  SMOOTH ABSOLUTE VALUE OF ONE OF THE EDGE FUNCTIONS

Taking the smooth absolute value

$$|x|_s = \begin{cases} \frac{\lambda}{2}x^2, & \text{if } |x| < \frac{1}{\lambda} \\ |x| - \frac{1}{2\lambda}, & \text{otherwise} \end{cases} \tag{22}$$

of $g_{abo}^{(r)}(\mathbf{r})$ resulted in slightly improved performance compared to keeping the function unmodified, see Appendix C.1. The factor $\lambda$ steers the scale of the smooth absolute value function and is here chosen as $\lambda = 1000$. The density is then given by $\hat{\rho}(\mathbf{r}) = \sum_{(a,b)\in\mathcal{E}_e}\sum_o^{N^o} g_{abo}^{(l)}(\mathbf{r})\left|g_{abo}^{(r)}(\mathbf{r})\right|_s +$ $\sum_{a\in\mathcal{N}}\hat{g}_a^{(l)}(\mathbf{r})\left|\hat{g}_a^{(r)}(\mathbf{r})\right|_s$.

## C  ABLATION STUDIES

### C.1  RADIAL CORRECTION AND SMOOTH ABSOLUTE VALUE

An ablation study of the radial correction factor and taking the smooth absolute value of one of the basis functions in each pair in the expansion of the density is performed. The results are shown in Table 3. For each configuration, three training runs are performed and the mean and standard error are reported. Both the radial correction and taking the absolute value significantly improve the performance, with the radial correction having the larger effect. The best performance is achieved when both techniques are used together.

Table 3: **Ablation study of taking the absolute value of one of the basis functions in the pair and applying the radial correction.** The small BOA version is trained on the QM9 VASP dataset. Errors are reported as NMAE [%].

| Absolute Value | Radial Correction | NMAE [%] |
|:---:|:---:|:---:|
| ✗ | ✗ | $0.204 \pm 0.002$ |
| ✓ | ✗ | $0.167 \pm 0.004$ |
| ✗ | ✓ | $0.1423 \pm 0.0006$ |
| ✓ | ✓ | $\mathbf{0.1381 \pm 0.0003}$ |

### C.2  CHOICE OF BASIS SET

The impact of the choice of basis sets is studied on two additional smaller basis sets, `def2-SVP` and `def2-TZVP` (Weigend & Ahlrichs, 2005). All hyperparameters other than the used basis are chosen as in BOA small and the models are trained on the QM9 VASP data. The results are shown in Table 4. Smaller basis sets still perform reasonably well, with the medium size basis `def2-TZVP` still surpassing the previous state of the art. Still, using larger basis sets improves the NMAE significantly.

Table 4: **Basis set ablation.** All settings other than the basis set are kept constant to the small BOA version and the models are trained on the QM9 VASP dataset. Three training runs are performed for each configuration and the mean and standard error are reported.

|            | NMAE [%]            | Number of basis functions |
|------------|---------------------|---------------------------|
| def2-SVP   | $0.194 \pm 0.001$   | 103                       |
| def2-TZVP  | $0.1504 \pm 0.0003$ | 192                       |
| def2-QZVPPD | $0.1381 \pm 0.0003$ | 374                       |

## C.3  QUADRATIC EXPANSION

To study the impact of the quadratic expansion we train additional models using the same hyperparameters as BOA small. When evaluating the density on the grid we drop $\hat{g}_a^{(r)}(\mathbf{r})$ and $g_{abo}^{(r)}(\mathbf{r})$ and expand the density linearly as $\rho(\mathbf{r}) = \sum_{a \in \mathcal{N}} \hat{g}_a^{(l)}(\mathbf{r}) + \sum_{(a,b) \in \mathcal{E}_e} \sum_o^{N^o} g_{abo}^{(l)}(\mathbf{r}) = \sum_{a \in \mathcal{N}} \sum_\mu \hat{g}_{a\mu}^{(l)} \omega_\mu^{Z_a}(\mathbf{r} - \mathbf{r}_a) + \sum_{(a,b) \in \mathcal{E}_e} \sum_o^{N^o} \sum_\mu g_{abo\mu}^{(l)} \omega_\mu^{Z_a}(\mathbf{r} - \mathbf{r}_a)$ where the sum over $b$ can be carried out before evaluating the basis functions $\omega_\mu^{Z_a}(\mathbf{r} - \mathbf{r}_a)$ on the grid. Three models were trained on the QM9 VASP data using this linear expansion and evaluated on the test set. The results are shown in Table 5. The quadratic expansion significantly outperforms the linear expansion, showing the importance of the quadratic expansion for accurate density predictions in BOA.

Table 5: **Quadratic expansion ablation.** All settings other than the expansion type are kept constant to the small BOA version and the models are trained on the QM9 VASP dataset. Three training runs are performed for each configuration and the mean and standard error are reported.

|                     | NMAE [%]            |
|---------------------|---------------------|
| Linear Expansion    | $0.2716 \pm 0.0007$ |
| Quadratic Expansion | $0.1381 \pm 0.0003$ |

## C.4  RADIAL CUTOFF

We conducted an additional experiment with larger cutoffs, specifically a message passing cutoff of $r_{mp} = 8$Å (instead of $r_{mp} = 6$Å) and an edge feature cutoff of $r_e = 4$Å (instead of $r_e = 3$Å) were chosen. Other than the cutoffs the BOA Small settings are used and three models are trained on the QM9 VASP data. Mean and standard error on the test set are reported in Table 6. Using larger cutoffs results in a small but consistent improvement in performance.

Table 6: **Radial cutoff ablation.** All settings other than the cutoffs are kept constant to the small BOA version and the models are trained on the QM9 VASP dataset. Three training runs are performed for each configuration and the mean and standard error are reported.

|                  | NMAE [%]            |
|------------------|---------------------|
| Standard cutoffs | $0.1381 \pm 0.0003$ |
| Larger cutoffs   | $0.1343 \pm 0.0007$ |

## D  DFT INITIALIZATION AND COULOMB ENERGY ERROR

To evaluate the quality of the predicted electron densities on metrics closer to practical applications, we calculate the Coulomb energy error and investigate the savings in KS-DFT calculations when using the predicted densities as initialization. Both of these evaluations are performed on the full QM9

PySCF test set of 10000 molecules using a BOA large model. BOA achieves a mean absolute error of 66 meV between the Coulomb energy calculated on the predicted densities and the converged label densities, comparing favorably to the 167 meV reported for the ResNet model (Li et al., 2025).

KS-DFT calculations using our densities as initial guesses and using a standard superposition of atomic densities as initial guesses are performed. The same settings as in the data generation described by Li et al. (2025) are used. With these settings the BOA initialization reduces the mean number of SCF iterations needed from 15.7 to 10.2, a reduction by 35%. This is a result competitive with the 35% reduction reported by Li et al. (2025) on the same data.

## E  EFFICIENCY

We evaluate the efficiency of ELECTRA Elsborg et al. (2025), SCDP Fu et al. (2024), standard BOA, and BOA with `def2-TZVP` basis models. We note that there is no difference in inference efficiency between the small and large settings, since only the number of probe points and the batch size during training are changed, and we therefore do not need to differentiate between them here. All models were evaluated on a 40GB A100 GPU using a block size maximizing the VRAM usage. The time per molecule is evaluated over the whole QM9 Vasp test set and the mean and standard deviation over these molecules are reported in Table 7. While BOA is roughly 2 times slower than SCDP, using the smaller `def2-TZVP` mitigates this effect to some extent and achieves a smaller error than the previous state of the art with efficiency comparable to SCDP. The efficiency/accuracy trade-off is easily steerable in BOA by employing basis sets of different size.

Table 7: **Time Efficiency.** The time per molecule needed during inference is evaluated on the QM9 VASP test set for ELECTRA (Elsborg et al., 2025), SCDP (Fu et al., 2024), standard BOA and BOA with `def2-TZVP` basis. The mean and standard deviation over all molecules in the test set are reported.

| Model | Time per molecule [s] |
| --- | --- |
| ELECTRA | $0.14 \pm 0.03$ |
| SCDP | $0.58 \pm 0.16$ |
| BOA | $1.27 \pm 0.27$ |
| BOA `def2-TZVP` | $0.64 \pm 0.15$ |

## F  ERROR DISTRIBUTION

The distribution of the electron density prediction errors on QM9 VASP is analyzed in more detail. Figures 5A and 5B show 2D histograms of the absolute errors of the predicted density for BOA large and SCDP (Fu et al., 2024), binned by the distance to the nearest and second-nearest atom. In each bin the absolute errors are summed. The errors are evaluated on 100 randomly sampled molecules and normalized by the total error of the SCDP model over all evaluated grid points. Additionally, Figure 5C shows the label electron density distribution binned in the same way and normalized by the total density over all evaluated grid points. Both models show the largest errors in the regions with high electron density. In these critical regions with the nearest atom at a distance of less than approximately $1.0$Å and the second-nearest atom at a distance of $0.5$Å to $1.5$Å, the BOA model, however, shows significantly smaller errors than SCDP.

## G  FUNCTION MESSAGE PASSING DERIVATION

We show that the messages $m_{abm\mu}$ are the coefficients in the basis of node $a$ that represent the feature functions $h_{bm}(\mathbf{r})$ of node $b$ best in the least squares sense. The message passing step therefore solves

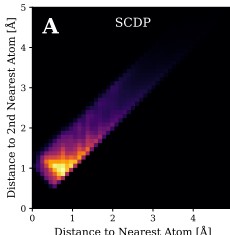 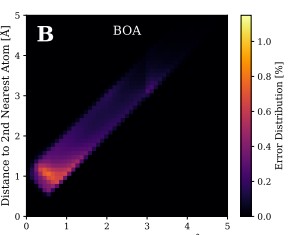 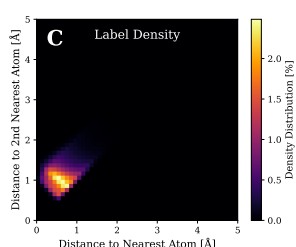

Figure 5: **Error distribution analysis.** The distribution of the absolute errors of the predicted electron density on the QM9 VASP dataset is shown as a function of the distance to the nearest and second-nearest atom. The errors are evaluated on 100 randomly sampled molecules and normalized by the total error of the SCDP model (Fu et al., 2024). **(A)** shows the error distribution of BOA large, **(B)** shows the error distribution of SCDP, and **(C)** shows the label electron density distribution, normalized by the total density over all evaluated grid points. Both models show the largest errors in the regions with high electron density, but BOA shows significantly smaller errors in these critical regions.

the optimization problem

$$\min_{m_{abm\mu}} \left\| h_{bm}(\mathbf{r}) - \sum_{\mu} m_{abm\mu} \omega_{\mu}^{Z_a}(\mathbf{r} - \mathbf{r}_a) \right\|^2 \tag{23}$$

$$\Rightarrow \min_{m_{abm\mu}} \left\| \sum_{\mu} h_{bm\mu} \omega_{\mu}^{Z_b}(\mathbf{r} - \mathbf{r}_b) - \sum_{\mu} m_{abm\mu} \omega_{\mu}^{Z_a}(\mathbf{r} - \mathbf{r}_a) \right\|^2. \tag{24}$$

Taking the derivative with respect to $m_{abm\mu}$ and setting it to zero results in

$$\int d\mathbf{r} \sum_{\mu} h_{bm\mu} \omega_{\mu}^{Z_b}(\mathbf{r} - \mathbf{r}_b) \omega_{\nu}^{Z_a}(\mathbf{r} - \mathbf{r}_a) = \int d\mathbf{r} \sum_{\mu} m_{abm\mu} \omega_{\mu}^{Z_a}(\mathbf{r} - \mathbf{r}_a) \omega_{\nu}^{Z_a}(\mathbf{r} - \mathbf{r}_a) \tag{25}$$

$$\Rightarrow \sum_{\mu} W_{\nu\mu}^{ab} h_{bm\mu} = \sum_{\mu} W_{\nu\mu}^{aa} m_{abm\mu} \tag{26}$$

$$\Rightarrow m_{abm\mu} = \sum_{\nu} (W^{aa})_{\mu\nu}^{-1} \sum_{\kappa} W_{\nu\kappa}^{ab} h_{bm\kappa}, \tag{27}$$

which is the solution to the optimization problem in Eq. 24 and exactly what is implemented in the message passing step.

## H  EQUIVARIANCE OF BOA

To understand the equivariance of the different BOA layers, it is necessary to understand how the basis functions $\omega_{\mu}^{Z}(\mathbf{r})$ transform under rotations. The angular part of the basis functions is given by spherical harmonics, which are known to transform under irreducible representations of the rotation group SO(3). These irreducible representations are indexed by angular momentum $l \in \mathbb{N}_0$ with dimension $2l + 1$. The basis functions can now be grouped according to their $l$ value into spherical tensors, i.e. there are groups of $2l + 1$ basis functions that transform under the irreducible representation indexed by $l$. We now relabel the basis functions by $i$ and $m \in \{-l_i, \dots, l_i\}$, where a basis function $\omega_m^i(\mathbf{r})$ is part of the $i$th tensor which transforms according to the representation indexed by $l_i$. The transformation under a rotation $\mathbf{R} \in SO(3)$ is given by

$$\omega_m^i(\mathbf{R}\mathbf{r}) = \sum_{m'} D_{mm'}^{l_i}(\mathbf{R}) \omega_{m'}^i(\mathbf{r}), \tag{28}$$

where $D_{mm'}^{l_i}(\mathbf{R})$ is the Wigner D-matrix. A set of basis functions $\omega_\mu^Z(\mathbf{r})$ can therefore be separated into irreducible representations

$$\boldsymbol{\omega}^Z(\mathbf{r}) = \left( \underbrace{\omega_{-l_1}^1(\mathbf{r}), \ldots, \omega_{l_1}^1(\mathbf{r})}_{2l_1+1}, \underbrace{\omega_{-l_2}^2(\mathbf{r}), \ldots, \omega_{l_2}^2(\mathbf{r})}_{2l_2+1}, \ldots \right)^T, \tag{29}$$

resulting in a block diagonal transformation behavior under rotations

$$\boldsymbol{\omega}^Z(\mathbf{R}\mathbf{r}) = \mathbf{D}^Z(\mathbf{R})\boldsymbol{\omega}^Z(\mathbf{r}), \tag{30}$$

where $\mathbf{D}^Z(\mathbf{R})$ is a block diagonal matrix with blocks given by the Wigner D-matrices corresponding to the $l$ values of the basis functions. The Wigner D-matrices are orthogonal matrices, so that $\mathbf{D}^Z(\mathbf{R})$ is also an orthogonal matrix, i.e. $\left(\mathbf{D}^Z(\mathbf{R})\right)^T = \left(\mathbf{D}^Z(\mathbf{R})\right)^{-1}$.

BOA employs the overlap and Coulomb matrices between the basis functions of different nodes, which are defined as

$$W_{\mu\nu}^{ab} = \int d\mathbf{r} \; \omega_\mu^{Z_a}(\mathbf{r} - \mathbf{r}_a)\omega_\nu^{Z_b}(\mathbf{r} - \mathbf{r}_b), \tag{31}$$

$$C_{\mu\nu}^{ab} = \int d\mathbf{r} \int d\mathbf{r}' \; \frac{\omega_\mu^{Z_a}(\mathbf{r} - \mathbf{r}_a)\omega_\nu^{Z_b}(\mathbf{r}' - \mathbf{r}_b)}{\|\mathbf{r} - \mathbf{r}'\|}. \tag{32}$$

Considering a rotation $\mathbf{R}$ of the molecule geometry, without changing the orientation of the basis functions, the overlap matrix transforms as

$$\bar{W}_{\mu\nu}^{ab} = \int d\mathbf{r} \; \omega_\mu^{Z_a}(\mathbf{r} - \mathbf{R}\mathbf{r}_a)\omega_\nu^{Z_b}(\mathbf{r} - \mathbf{R}\mathbf{r}_b) = \int d\mathbf{r}' \; \omega_\mu^{Z_a}(\mathbf{R}\mathbf{r}' - \mathbf{R}\mathbf{r}_a)\omega_\nu^{Z_b}(\mathbf{R}\mathbf{r}' - \mathbf{R}\mathbf{r}_b) \tag{33}$$

$$= \int d\mathbf{r}' \; \sum_{\mu'} D_{\mu\mu'}^{Z_a}(\mathbf{R})\omega_{\mu'}^{Z_a}(\mathbf{r}' - \mathbf{r}_a) \sum_{\nu'} D_{\nu\nu'}^{Z_b}(\mathbf{R})\omega_{\nu'}^{Z_b}(\mathbf{r}' - \mathbf{r}_b) \tag{34}$$

$$= \sum_{\mu'} D_{\mu\mu'}^{Z_a}(\mathbf{R}) \sum_{\nu'} D_{\nu\nu'}^{Z_b}(\mathbf{R}) \int d\mathbf{r}' \; \omega_{\mu'}^{Z_a}(\mathbf{r}' - \mathbf{r}_a)\omega_{\nu'}^{Z_b}(\mathbf{r}' - \mathbf{r}_b) \tag{35}$$

$$= \sum_{\mu'} D_{\mu\mu'}^{Z_a}(\mathbf{R}) \sum_{\nu'} D_{\nu\nu'}^{Z_b}(\mathbf{R}) W_{\mu'\nu'}^{ab}. \tag{36}$$

$$\Rightarrow \bar{\mathbf{W}}^{ab} = \mathbf{D}^{Z_a}(\mathbf{R})\mathbf{W}^{ab}\left(\mathbf{D}^{Z_b}(\mathbf{R})\right)^T, \tag{37}$$

where $\bar{W}$ is the transformed matrix. This notation will be used throughout this section. Completely analogously, the Coulomb matrix transforms

$$\bar{\mathbf{C}}^{ab} = \mathbf{D}^{Z_a}(\mathbf{R})\mathbf{C}^{ab}\left(\mathbf{D}^{Z_b}(\mathbf{R})\right)^T. \tag{38}$$

Similarly, one can consider a function $f(\mathbf{r}) = \sum_\mu f_\mu \omega_\mu^Z(\mathbf{r})$. The rotated function $\bar{f}(\mathbf{r})$ represented in the same basis with rotated molecule geometry is given by

$$\bar{f}(\mathbf{r}) = f(\mathbf{R}^{-1}\mathbf{r}) = \sum_\mu f_\mu \omega_\mu^Z(\mathbf{R}^{-1}\mathbf{r}) = \sum_\mu f_\mu \sum_{\mu'} D_{\mu\mu'}^Z(\mathbf{R}^{-1})\omega_{\mu'}^Z(\mathbf{r}) = \sum_{\mu'} \bar{f}_{\mu'}\omega_{\mu'}^Z(\mathbf{r}), \tag{39}$$

$$\Rightarrow \bar{f}_{\mu'} = \sum_\mu D_{\mu\mu'}^Z(\mathbf{R}^{-1})f_\mu. \tag{40}$$

$$\Rightarrow \bar{\mathbf{f}} = \left(\mathbf{D}^Z(\mathbf{R}^{-1})\right)^T \mathbf{f} = \mathbf{D}^Z(\mathbf{R})\mathbf{f}. \tag{41}$$

### H.1 MESSAGE PASSING

The overlap integrals $o_{abm\mu}$ defined in Eq. 9 transform as

$$\bar{o}_{abm\mu} = \sum_\nu \bar{W}_{\mu\nu}^{ab}\bar{h}_{bm\nu} = \sum_\nu \sum_{\mu'} D_{\mu\mu'}^{Z_a}(\mathbf{R}) \sum_{\nu'} D_{\nu\nu'}^{Z_b}(\mathbf{R}) W_{\mu'\nu'}^{ab} \sum_{\nu''} D_{\nu\nu''}^{Z_b}(\mathbf{R}) h_{bm\nu''} \tag{42}$$

$$= \sum_{\mu'} D_{\mu\mu'}^{Z_a}(\mathbf{R}) o_{abm\mu'}, \tag{43}$$

leading to the transformation behavior

$$\bar{m}_{abm\mu} = \sum_{\nu} (\bar{W}^{aa})^{-1}_{\mu\nu} \bar{o}_{abm\nu} = \sum_{\nu} \sum_{\nu'} \sum_{\mu'} \sum_{\nu''} D^{Z_a}_{\mu\mu'}(\mathbf{R})(W^{aa})^{-1}_{\mu'\nu'} D^{Z_a}_{\nu\nu'}(\mathbf{R}) D^{Z_a}_{\nu\nu''}(\mathbf{R}) o_{abm\nu''}$$
(44)

$$= \sum_{\mu'} D^{Z_a}_{\mu\mu'}(\mathbf{R}) m_{abm\mu'}$$
(45)

of the messages $m_{abm\mu}$. The overlap integrals $\alpha_{abmn}$ are scalars, because

$$\bar{\alpha}_{abmn} = \sum_{\mu\nu} \bar{h}_{am\mu} \bar{W}^{ab}_{\mu\nu} \bar{h}_{bn\nu}$$
(46)

$$= \sum_{\mu\nu} \sum_{\mu'} D^{Z_a}_{\mu\mu'}(\mathbf{R}) h_{am\mu'} \sum_{\mu''} D^{Z_a}_{\mu\mu''}(\mathbf{R}) \sum_{\nu'} D^{Z_b}_{\nu\nu'}(\mathbf{R}) W^{ab}_{\mu''\nu'} \sum_{\nu''} D^{Z_b}_{\nu\nu''}(\mathbf{R}) h_{bn\nu''}$$
(47)

$$= \sum_{\mu'} h_{am\mu'} \sum_{\nu'} W^{ab}_{\mu'\nu'} h_{bn\nu'} = \alpha_{abmn}.$$
(48)

For scalar features any MLP is equivariant, since scalars are invariant under rotations. The attention weights $\tilde{\alpha}_{abmn}$ are therefore also scalars. The updated node features $\tilde{h}_{am\mu}$ transform as

$$\bar{\tilde{h}}_{am\mu} = \sum_{b \in \mathcal{N}_{\text{mp}}(a)} \sum_{n} \bar{\tilde{\alpha}}_{abmn} \bar{m}_{abn\mu}$$
(49)

$$= \sum_{b \in \mathcal{N}_{\text{mp}}(a)} \sum_{n} \tilde{\alpha}_{abmn} \sum_{\mu'} D^{Z_a}_{\mu\mu'}(\mathbf{R}) m_{abn\mu'}$$
(50)

$$= \sum_{\mu'} D^{Z_a}_{\mu\mu'}(\mathbf{R}) \tilde{h}_{am\mu'},$$
(51)

showing that the message passing layer is equivariant.

## H.2 Nonlinearity

The nonlinearity is based on calculating scalar quantities, which are then transformed by an MLP, and weighting the original features with these scalars. The scalar features $l_{amn}$ are invariant under rotations because

$$\bar{l}_{amn} = \sum_{\mu\nu} \bar{h}_{am\mu} \bar{C}^{aa}_{\mu\nu} \bar{h}_{an\nu}$$
(52)

$$= \sum_{\mu\nu} \sum_{\mu'} D^{Z_a}_{\mu\mu'}(\mathbf{R}) h_{am\mu'} \sum_{\mu''} D^{Z_a}_{\mu\mu''}(\mathbf{R}) \sum_{\nu'} D^{Z_a}_{\nu\nu'}(\mathbf{R}) C^{aa}_{\mu''\nu'} \sum_{\nu''} D^{Z_a}_{\nu\nu''}(\mathbf{R}) h_{an\nu''}$$
(53)

$$= \sum_{\mu'} h_{am\mu'} \sum_{\nu'} C^{aa}_{\mu'\nu'} h_{an\nu'} = l_{amn}.$$
(54)

Therefore, the weights $w_{amn}$ are also scalars and the updated node features $\tilde{h}_{am\mu}$ transform as

$$\bar{\tilde{h}}_{am\mu} = \sum_{n} \bar{w}_{amn} \bar{h}_{an\mu} = \sum_{n} w_{amn} \sum_{\mu'} D^{Z_a}_{\mu\mu'}(\mathbf{R}) h_{an\mu'}$$
(55)

$$= \sum_{\mu'} D^{Z_a}_{\mu\mu'}(\mathbf{R}) \tilde{h}_{am\mu'},$$
(56)

showing that the nonlinearity is equivariant.

## H.3 L2 Normalization

The L2 normalization of the features is done per node and feature channel, i.e. for each $a$ and $m$. The normalization factor is a scalar because

$$\bar{n}_{am} = \sqrt{\sum_{\mu\nu} \bar{h}_{am\mu} \bar{W}^{aa}_{\mu\nu} \bar{h}_{am\nu}} \tag{57}$$

$$= \sqrt{\sum_{\mu\nu} \sum_{\mu'} D^{Z_a}_{\mu\mu'}(\mathbf{R}) h_{am\mu'} \sum_{\mu''} D^{Z_a}_{\mu\mu''}(\mathbf{R}) \sum_{\nu'} D^{Z_a}_{\nu\nu'}(\mathbf{R}) W^{aa}_{\mu''\nu'} \sum_{\nu''} D^{Z_a}_{\nu\nu''}(\mathbf{R}) h_{am\nu''}} \tag{58}$$

$$= \sqrt{\sum_{\mu'} h_{am\mu'} \sum_{\nu'} W^{aa}_{\mu'\nu'} h_{am\nu'}} = n_{am}. \tag{59}$$

Therefore, the normalized features $\tilde{h}_{am\mu}$ are straightforwardly equivariant.

## H.4 Edge Update

Exactly the same arguments as for the L2 normalization and the nonlinearity can be used to show that $o^{(n)}_{abmn}$ and $o^{(e)}_{abmn}$ are scalars. Therefore, also the generated weights $s^{(\cdot,\cdot)}_{ab}$ and $\tilde{w}^{(\cdot,\cdot)}_{abmn}$ are scalars, making the update step in Eq. 21 equivariant.

In summary, all operations in BOA transform equivariantly under rotations, so that the whole model is SO(3)-equivariant.

# I Hyperparameters and Training Details

The model is trained in two stages, the pre-training of the initial guess followed by the full training. During training an exponential moving average (EMA) of the previous weights is used. For testing the epoch that performed best in validation is chosen in all experiments. A mean absolute error loss on the electron density values on the grid is used. When evaluating the density on the grid, edge function values are set to 0 if both of the edge functions have a distance of more than 3 Å from the grid point. The electron density prediction on the grid is multiplied with the standard deviation of the labels calculated over the whole training set. The predicted density is additionally multiplied with 0.1, as this stabilized training in initial experiments.

## I.1 Model Hyperparameters

The detailed list of hyperparameters of the model is given in Table 8. Settings changed for the pre-training of the initial guess are listed in Table 9. One special layer is the MLP for the edge update, which consists of a two linear layers, with first a SiLU activation and then a LayerNorm between them. The first layer has $2 \cdot (N^c)^2$ input and output neurons, and the second has $4 \cdot N^c \cdot (N^c + 1)$ output neurons.

The MLP used in the nonlinearity consists of two linear layers with $(N^c)^2$ input and output neurons, separated by first a SiLU activation and then a LayerNorm.

The Gaussian radial embedding employed in the radial correction uses 50 Gaussians evenly spaced between $0a_0$ and $3a_0$ (with $a_0$ being the Bohr radius). Therefore, 50 values are fed into the following MLP, which consists of first a linear layer with dimension $50 \times 50$ then SiLU activation and a linear layer with dimension $50 \times N^{\exp}$. Here $N^{\exp}$ is the number of different exponents used in the radial parts of the basis functions, i.e. the number of unique radial functions. For basis functions belonging to the same irreducible representations of SO(3), and therefore having the same radial part, the same correction factor is used.

**Differences between BOA sizes** Going from the small to the large version of BOA, the number of randomly probed points per molecule is increased from 5000 to 6000. Additionally, the batch size is increased from 12 to 24 which also leads to an increase in "epochs", since the maximum number of steps remains at 500000.

Table 8: The hyperparameters of BOA on the QM9 dataset using the small settings.

| Hyperparameter | Value |
|---|---|
| Batch Size | 12 |
| Initial Learning Rate | 0.001 |
| Learning Rate Schedule | CosineAnnealingLR |
| $T_{\max}$ | 500000 steps |
| Optimizer | Adam |
| $\beta_1, \beta_2, \epsilon$ | $(0.9, 0.999, 10^{-8})$ |
| Weight Decay | 0.0 |
| Max Training Steps | 500000 |
| Gradient Clipping Value | 0.5 |
| EMA Decay | 0.995 |
| # BOA blocks | 4 |
| # hidden channels $N^c$ | 32 |
| # edge functions after partial channel mean | 8 |
| MLP activation | SiLU |
| Initial Guess Channels | 1 |
| Absolute Value Scale $\lambda$ | 1000 |
| scale factor | 0.1 |
| Probed Points | 5000 |
| Message passing cutoff $r_{mp}$ | 6 Å |
| Edge function cutoff $r_e$ | 3 Å |
| Basis Set | def2-QZVPPD |

Table 9: The pretraining hyperparameters that differ from the default settings in Table 8.

| Hyperparameter | Value |
|---|---|
| Initial Learning Rate | 0.001 |
| Learning Rate Scheduler | CosineAnnealingWarmRestarts |
| $T_0$ | 250 |
| $T_{\mathrm{mult}}$ | 1 |
| # Steps | 1000 |

**Differences between datasets** On the MD datasets, the number of maximum training steps was reduced to 200000.

## I.2 BASIS SET

The number of basis functions per atom type and shell of def2-QZVPPD basis set is given in Table 10.

Table 10: Uncontracted (primitive) shells per $\ell$ in def2-QZVPPD and total spherical functions.

| Element | 0 | 1 | 2 | 3 | 4 | Total (sph.) |
|---|---|---|---|---|---|---|
| H | 7 | 4 | 2 | 1 | 0 | 36 |
| C | 16 | 8 | 4 | 2 | 1 | 83 |
| N | 16 | 8 | 4 | 2 | 1 | 83 |
| O | 16 | 9 | 4 | 2 | 1 | 86 |
| F | 16 | 9 | 4 | 2 | 1 | 86 |

## J    LLM USAGE

Large Language Models (LLMs) were used to assist with writing (formulation and wording), coding (AI autocomplete), and gathering resources in the preparation of this article.

