# OpenReview forum: "A Function-Centric Graph Neural Network Approach for Predicting Electron Densities"
_ICLR.cc/2026/Conference — ICLR 2026 Poster_

### Official Review · Reviewer_1d9S · 2025-10-31

**Soundness:** 3
**Presentation:** 3
**Contribution:** 3
**Rating:** 6
**Confidence:** 2

**Summary:**

The authors build an equivariant architecture to perform electron density prediction using discretized bases, ultimately achieving significantly lower density prediction errors than previous work on small molecule datasets. The key innovation is in the message passing scheme used. The paper is generally well written, although there are some parts that I am still trying to understand (see the questions below).

**Strengths:**

The background work seems thorough (although I am personally only familiar with ELECTRA among them), and there is a clear improvement in electron density prediction.

**Weaknesses:**

The main body of the paper is methods-heavy and light on results. Certain components (eg, gated nonlinearities) are more well-known and could be shifted to the appendix. More explanatory figures/schematics could be very useful.

**Questions:**

Is there a benefit to predicting the electron density in this basis over learning the elements of the density matrix, similarly to what's done in Hamiltonian prediction models? Wouldn't this avoid going through the intermediate grid representation?

What is the complexity of the forward pass? For the second class of 'volumetric' prediction methods, I imagine the scalability to larger structures is quite poor. How much does use of these discrete methods improve on that?

The basis sets used are quite large, and if I remember correctly, QM9 uses HCONF elements. Do the authors expect that this could eventually scale well to heavy elements where the number of basis functions would become even larger?

What are the green circles in Fig. 1c? I looked at this subfigure for a while, since it seemed to be the 'visual explanation' that I was looking for, but am still confused.

How sensitive is the model to radial cutoffs? From what I've seen, MLIPs typically use smaller cutoffs (4-6 A) and Hamiltonian prediction models tend to use larger ones. Since the electron density is a more 'fundamental' quantity directly computed from Hamiltonian and density matrices, I'm wondering if electron density prediction models similarly benefit from larger cutoffs.

Can the authors discuss the error metric a bit more in detail? They use NMAE, which weights all grid points equally. How is the error usually distributed among grid points? Is it concentrated in areas where the electron density has a higher spatial variation? How does the distribution of the error vary between models?

---

> ### Author Response · Authors · 2025-11-22
> **Answer to Reviewer 1d9S**
>
> Thank you for your constructive and helpful comments and questions. We are happy to hear that you appreciate the improvement BOA offers in electron density prediction. We address your comments and questions in the following.
>
> ### Additional Results
> >The main body of the paper is methods-heavy and light on results. Certain components (eg, gated nonlinearities) are more well-known and could be shifted to the appendix. More explanatory figures/schematics could be very useful.
>
> We agree that the submission is heavy on methods, but we believe our core contributions to be the methods justifying spending most of the main body of the submission on these. To offer more practical demonstrations, we worked to add additional experiments both to the main text and to the appendix in the form of 1. an analysis on the generalization to larger molecules, 2. an evaluation of the predicted electron densities on the Coulomb energy, 3. an evaluation of the predicted electron densities as initialization for DFT calculations, 4. an ablation on the choice of basis sets, 5. an ablation of the quadratic expansion used in BOA, 6. an experiment evaluating the impact of choosing larger cutoffs, 7. an efficiency analysis, and 8. an analysis of the error distribution. We hope this addresses the concerns about too few results.
>
> ### Predicting the Density Matrix Directly
> >Is there a benefit to predicting the electron density in this basis over learning the elements of the density matrix, similarly to what's done in Hamiltonian prediction models? Wouldn't this avoid going through the intermediate grid representation?
>
> We appreciate that your proposal would allow avoiding evaluation on the grid, which is attractive, since a large amount of the needed compute goes to this evaluation. We however see benefits in evaluating the density on the grid, as is done in this work. While the evaluation on the grid is costly, it ensures that the used loss measures what we actually care about: how close the predicted ground state electron density is to the ground truth label. Comparing the elements of the density matrix could be more problematic, since there are representations of the electron density that are close in function space, but far in coefficient space, because the basis of products of basis functions is highly non-orthogonal. In this sense comparing densities on the grid leaves more freedom for the model to choose the learned coefficients.
>
> Additionally, the density matrix also fulfills properties, i.e., is low-rank and idempotent (with the overlap matrix as metric), that are not enforced in the representation learned by BOA. In BOA only the blocks of the density matrix learned per edge are low-rank, differing from the density matrix, which is "fully" low-rank. Not enforcing these properties leaves more freedom for BOA, making the model more expressive. The radial correction and smooth absolute value used in the best BOA models are also only possible when evaluating on the grid.
>
> ### Scaling to Larger Molecules
> >What is the complexity of the forward pass? For the second class of 'volumetric' prediction methods, I imagine the scalability to larger structures is quite poor. How much does use of these discrete methods improve on that?
>
> See the **Scalability** Section in the answer to Reviewer *igug*, who asked a similar question.
>
> ### Scaling to Heavier Elements
> >The basis sets used are quite large, and if I remember correctly, QM9 uses HCONF elements. Do the authors expect that this could eventually scale well to heavy elements where the number of basis functions would become even larger?
>
> It is indeed true that scaling to a higher number of basis function can pose a challenge. Since the datasets evaluated here are generated employing pseudo-potentials and the learned electron densities only contain valence electrons, we do, however, expect it to be feasible to switch to a basis set optimized for use in combination with pseudo-potentials, e.g. *GTH-TZV2P*. These basis sets contain a much smaller number of basis functions for heavy elements and should make scaling less challenging. We perform an additional experiment to confirm the feasibility of *GTH-TZV2P*:
> |  | NMAE [%] | Number of basis functions for QM9 |  Number of basis functions for Cl |
> |:-----------|:-------------------:|:-------------------:|:-------------------:|
> | GTH-TZV2P | 0.1539 ± 0.0009 | 131 | 30 |
> | def2-QZVPPD | 0.1381 ± 0.0003 | 374 | 114 |
>
> Using this smaller basis BOA still achieves an error smaller than the previous state of the art. We also report the number of basis functions needed for QM9 (with H, C, O, N, F) and the additional basis functions needed to include chlorine. The number of basis functions needed for chlorine using *GTH-TZV2P* is not larger than for lighter atoms, while it is much larger for *def2-QZVPPD*.

---

> ### Author Response · Authors · 2025-11-23
> **Answer to Reviewer 1d9S (continued)**
>
> ### Clarification of Figure 1C
> >What are the green circles in Fig. 1c? I looked at this subfigure for a while, since it seemed to be the 'visual explanation' that I was looking for, but am still confused.
>
> The green circles in Figure 1C show the centers of basis function pairs $\omega^{Z\_a}\_\mu(\mathbf{r} - \mathbf{r}_a)\omega^{Z\_b}\_\mu(\mathbf{r} - \mathbf{r}\_b)$ that show up in the expansion of the electron density used in BOA. The radial part of a product of two Gaussian type basis functions contains the product of two Gaussian functions, each of which are centered at one of the atoms. This product can be rewritten to form a Gaussian centered on the edge connecting the two atoms. These centers are shown as green circles in Figure 1C. We thank the Reviewer for bringing to our attention that this is not adequately explained in the figure caption; we added further clarification in the submission.
>
> ### Larger Radial Cutoffs
> >How sensitive is the model to radial cutoffs? From what I've seen, MLIPs typically use smaller cutoffs (4-6 A) and Hamiltonian prediction models tend to use larger ones. Since the electron density is a more 'fundamental' quantity directly computed from Hamiltonian and density matrices, I'm wondering if electron density prediction models similarly benefit from larger cutoffs.
>
> We thank the Reviewer for the suggestion to investigate how performance increases when using a larger cutoff. We conducted an additional experiment with larger cutoffs, specifically a message passing cutoff of $r\_{mp} = 8$Å (instead of $r\_{mp} = 6$Å) and an edge feature cutoff of $r\_{e} = 4$Å (instead of $r\_{e} = 3$Å) were chosen. Other than the cutoffs, the *BOA Small* settings are used and three models are trained on the QM9 VASP data. Mean and standard error on the test set are reported:
>
> |  | NMAE [%] |
> |:-----------|:-------------------:|
> | Standard cutoffs  | 0.1381 ± 0.0003 |
> | Large cutoffs | 0.1343 ± 0.0007 |
>
> A slight improvement is achieved by choosing the larger cutoffs. We note however, that this comes at significant cost considering efficiency, especially because of the larger edge feature cutoff $r\_{e}$ that leads to a larger number of basis functions that need to be evaluated on the grid.
>
> ### Error Distribution
> >Can the authors discuss the error metric a bit more in detail? They use NMAE, which weights all grid points equally. How is the error usually distributed among grid points? Is it concentrated in areas where the electron density has a higher spatial variation? How does the distribution of the error vary between models?
>
> The distribution of the electron density prediction errors on QM9 VASP is analyzed in more detail. Figure 5A and 5B in Appendix A.4 of the revised submission show 2D histograms of the absolute errors of the predicted density for *BOA large* and SCDP [2], binned by the distance to the nearest and second-nearest atom. In each bin the absolute errors are summed. The errors are evaluated on 100 randomly sampled molecules and normalized by the total error of the SCDP model over all evaluated grid points. Additionally, Figure 5C shows the label electron density distribution binned in the same way and normalized by the total density over all evaluated grid points. Both models show the largest errors in the regions with high electron density. In these critical regions with the nearest atom at a distance of less than roughly $1.0$Å and the second-nearest atom at a distance of $0.5$Å to $1.5$Å, the BOA model however shows significantly smaller errors than SCDP.
>
> We hope that we were able to address all your questions and comments and are happy to answer any additional questions you may have.
>
> [1] Li, Chenghan, Or Sharir, Shunyue Yuan, and Garnet Kin-Lic Chan. “Image Super-Resolution Inspired Electron Density Prediction.” Nature Communications 16, no. 1 (2025): 4811. https://doi.org/10.1038/s41467-025-60095-8.
>
> [2] Fu, Xiang, Andrew Scott Rosen, Kyle Bystrom, et al. “A Recipe for Charge Density Prediction.” Paper presented at The Thirty-eighth Annual Conference on Neural Information Processing Systems. November 6, 2024. https://openreview.net/forum?id=b7REKaNUTv.

---

> > ### Author Response · Authors · 2025-12-03
> > **Answer to Reviewer 1d9S (continued)**
> >
> > ### Additional Figure Illustrating Basis Overlap Message Passing
> > > More explanatory figures/schematics could be very useful.
> >
> > To further illustrate the message passing procedure introduced in this work, we added Figure 2 in the revised manuscript, showing the basis overlap message passing scheme for a one-dimensional example.

---

### Official Review · Reviewer_TU9J · 2025-10-31

**Soundness:** 2
**Presentation:** 1
**Contribution:** 3
**Rating:** 4
**Confidence:** 3

**Summary:**

The paper introduces Basis Overlap Architecture, an equivariant GNN-based model to predict ground-state electron densities. The direct density prediction aims to replace the computationally expensive KD-DFT calculations. The authors develop an innovative message scheme that casts internal network features as basis functions and computes an overlap matrix of interacting vertices to compute messages. Instead of a linear expansion, the model casts the density as a quadratic expansion of the learned basis functions. Attention is calculated based on the overlap matrix and Coulomb matrix for additional physical inductive bias. BOA shows strong performance on density prediction on the QM9 and MD datasets.

**Strengths:**

1. The internal density representation and the model architecture are well motivated, have the proper inductive bias. The problem of fast density generation is important for molecule discovery and characterization
2. Representing internuclear regions without virtual nodes is a novel innovation

**Weaknesses:**

1.  "In contrast to previous work, we however, do not expand the density or its square root directly as" Lin3 165-166.  citations here would be very helpful to situate the work
2. Unclear presentation of Figure 1 and Figure 2. What is the MN block in Figure 2. The notation is difficult to follow. In general, the paper was difficult to follow and missing key information

**Questions:**

1. "Superscripts l and r denote left and right, respectively, for reasons that are clear from Equation 2" Line 155-156. It's not clear why the node features also have this directionality in the first term in Eq. 2.
2. "BOA uses an initial guess based on the atom types of the nodes." Line 199- Initial guess of the coefficients for the atom features?
3. How is the model trained? What loss function is used?

---

> ### Author Response · Authors · 2025-11-22
> **Answer to Reviewer TU9J**
>
> Thank you for your review and helpful comments. We are happy to hear that you found the model architecture well motivated and innovative. We address your comments and questions in the following.
>
> ### Additional Citations
> > "In contrast to previous work, we however, do not expand the density or its square root directly as" Lin3 165-166. citations here would be very helpful to situate the work
>
> Thank you for helping situate this contribution! The respective references have now been added to the updated submission.
>
> ### Presentation
> > Unclear presentation of Figure 1 and Figure 2. What is the MN block in Figure 2. The notation is difficult to follow. In general, the paper was difficult to follow and missing key information
>
> We added further explanation to the caption of Figures 1 and 2 (Figures 1 and 3 in the revised submission). The MN block refers to the matrix normalization described in Section 2.8 on the edge update, specifically Equation (20). We thank the reviewer for bringing to our attention that this was not clarified in the caption, and we added it to the submission. Regarding the missing key information, we would be very happy to provide any information and include it in the submission. We have amended the presentation while respecting the page limit. If after these improvements any key information is still missing, we kindly ask for an explicit list of issues or questions that we will be happy to address in the remaining discussion phase or the manuscript itself.
>
> ### Superscripts *l* and *r* in Initial Guess
> >"Superscripts l and r denote left and right, respectively, for reasons that are clear from Equation 2" Line 155-156. It's not clear why the node features also have this directionality in the first term in Eq. 2.
>
> The initial guess functions $\hat{g}^{(l)}\_{a}$ and $\hat{g}^{(r)}\_{a}$ can be understood as edge features that exist on every self-loop in the graph. While the self-loop begins and ends at the same node, there are still two functions assigned to these edges. While the functions $\hat{g}^{(l)}\_{a}$ and $\hat{g}^{(r)}\_{a}$ can be assigned to a single node $a$ in the same sense in which a self-loop can be assigned to a single node, the expansion of the initial guess is still quadratic and two distinct feature functions are learned per node.
>
> ### Initial Guess
> >"BOA uses an initial guess based on the atom types of the nodes." Line 199- Initial guess of the coefficients for the atom features?
>
> In Section 2.2, we describe how the initial guess of the electron density is pre-trained. The coefficients $\hat{g}^{(l)}\_{a\mu}$ and $\hat{g}^{(r)}\_{a\mu}$ that represent the initial guess functions $\hat{g}^{(l)}\_{a}$ and $\hat{g}^{(r)}\_{a}$ are set based on the atom type of the node $a$. This line therefore does not refer to the initialization of the node or edge features, which is described in Section 2.3, but to the initial guess of the density. Only the difference between this initial guess and the label electron density is predicted by the model. We have updated the text to clarify this important point.
>
> ### Training Details
> >How is the model trained? What loss function is used?
>
> A mean absolute error is calculated on the predicted electron density on the grid and is used as a training loss. This information can be found in Appendix A.4 of the original submission. We have now changed the submission to more clearly refer to this section. Appendix A.4 (Appendix A.7 in the revised submission) also describes details of the BOA training. Please ask in case any specific information is still missing.
>
> We hope we were able to address the raised questions and are happy to answer any additional questions that you may have.

---

### Official Review · Reviewer_igug · 2025-10-31

**Soundness:** 3
**Presentation:** 3
**Contribution:** 3
**Rating:** 6
**Confidence:** 4

**Summary:**

This paper introduces Basis Overlap Architecture (BOA), a new equivariant message passing neural network for predicting ground-state electron densities. Unlike prior approaches that expand the density linearly in atom-centered basis functions, BOA represents it as a quadratic expansion, inspired by the density matrix representation in Kohn–Sham DFT. The model leverages basis-function overlaps to define message passing between nodes, effectively incorporating both molecular geometry and basis information into the communication scheme.

**Strengths:**

**Novel and physically grounded formulation** : The idea of treating node features as functions represented in a basis and defining message passing through basis overlaps is elegant and well-motivated.
This approach embeds physical inductive bias (basis overlap, geometry dependence) directly into the architecture, rather than as post-hoc constraints.

**Comprehensive evaluation**: Experiments on two major datasets (QM9–VASP and MD–DFT) demonstrate strong and consistent improvements. The comparisons to strong recent baselines (SCDP, ELECTRA, GPWNO) are appropriate and show clear quantitative gains.

**Weaknesses:**

**Lack of quantitative validation on physical observables**: While the model demonstrates impressive accuracy in predicting electron densities, the paper does not quantify how this improvement translates into physically meaningful quantities — such as total energy, dipole moments, or electrostatic potentials. Without showing how the predicted densities affect DFT-derived properties or the self-consistent field (SCF) convergence, the connection between the proposed method and its stated motivation (accelerating or improving DFT) remains incomplete.

**Ablation and efficiency analysis**: The paper lacks quantitative ablations demonstrating which design choices (e.g., basis choice like def2-SVP, basis-overlap attention, quadratic expansion, learned radial corrections) most contribute to the observed performance. The large model reportedly requires ∼94 GB GPU memory, which raises questions about scalability and computational efficiency.

**Minor**
- Eq. (13) and others: Consider to change $(a,b) \in \mathcal E$ to $b \in \mathcal N(a)$ or equivalent so that $a$ remains a free index.

**Questions:**

1. **(Efficiency)** Have you evaluated the computational efficiency of BOA in terms of speed and memory usage compared to ELECTRA or other recent electron-density models?

2. **(Basis dependence)** How sensitive is BOA’s performance to the choice of basis set (e.g., def2-SVP vs. def2-QZVPPD)? Have you observed any notable differences in accuracy or stability?

3. **(Non–atom-centric extensions)** Could BOA be extended to incorporate non–atom-centered or floating basis functions, similar to ELECTRA’s floating orbitals, to further enhance flexibility?

4. **(Relation to KS-DFT coefficients)** Do the predicted coefficients correspond directly to the Kohn–Sham orbital coefficient matrix, or are they only related to the density-matrix blocks derived from it?

5. **(Scalability)** What is the computational scaling behavior of BOA with respect to the number of atoms or basis functions? Have you tested the model’s scalability on larger molecular systems?

---

> ### Author Response · Authors · 2025-11-22
> **Answer to Reviewer igug**
>
> Thank you for your detailed review and constructive comments and questions. We appreciate that you found the new message passing formulation well-motivated and elegant. Thank you as well for your comments on additional ablations which we address in the following.
>
> ### Quantitative validation on physical observables
> > Lack of quantitative validation on physical observables: While the model demonstrates impressive accuracy in predicting electron densities, the paper does not quantify how this improvement translates into physically meaningful quantities — such as total energy, dipole moments, or electrostatic potentials. Without showing how the predicted densities affect DFT-derived properties or the self-consistent field (SCF) convergence, the connection between the proposed method and its stated motivation (accelerating or improving DFT) remains incomplete.
>
> We perform two additional evaluations to address this point, using the predicted electron densities as initialization for KS-DFT calculations and calculating the error in Coulomb energy of the predicted density.
>
> KS-DFT calculations using our densities as initial guesses and using a standard superposition of atomic densities as initial guesses are performed. The same settings as in the data generation described in [6] are used. With these settings, the BOA initialization reduces the mean number of SCF iterations needed from $15.7$ to $10.2$, a reduction by $35$%. This is a result competitive with the $35$% reduction reported in [6] on the same data.
>
> The Coulomb energies of the predicted electron densities and the converged ground state electron densities are calculated. BOA achieves a mean absolute error of $66$ meV, comparing favorably to the $167$ meV reported for the ResNet model [6].

---

> ### Author Response · Authors · 2025-11-22
> **Answer to Reviewer igug (continued)**
>
> ## Ablation and efficiency analysis
> >  The paper lacks quantitative ablations demonstrating which design choices (e.g., basis choice like def2-SVP, basis-overlap attention, quadratic expansion, learned radial corrections) most contribute to the observed performance.
>
> To further clarify which design choices most contribute to the performance of BOA, we performed additional ablation studies.
>
> ### Basis set choice
> > (Basis dependence) How sensitive is BOA’s performance to the choice of basis set (e.g., def2-SVP vs. def2-QZVPPD)? Have you observed any notable differences in accuracy or stability?
>
> The impact of the choice of basis sets is studied on two additional smaller basis sets, *def2-SVP* and *def2-TZVP*. All hyperparameters other than the used basis are chosen as in *BOA small* and the models are trained on the QM9 VASP data. The mean and standard error over three models with different seeds are reported:
>
> | Basis Set | NMAE [%] | Number of basis functions |
> |:-----------|:-------------------:|:------------:|
> | *def2-SVP*  | 0.194 ± 0.001     | 103           |
> | *def2-TZVP* | 0.1504 ± 0.0003   | 192           |
> | *def2-QZVPPD* | 0.1381 ± 0.0003 | 374           |
>
> Smaller basis sets still perform reasonably well, with the medium-sized basis *def2-TZVP* still surpassing the previous state of the art. Even so, using larger basis sets improves the NMAE significantly. Employing BOA using these other basis sets did not pose additional challenges. There was no additional hyperparameter tuning needed, training stability did not change, and these results were achieved out of the box.
>
> ### Quadratic Expansion
> To study the impact of the quadratic expansion, we trained additional models using the same hyperparameters as *BOA small*. When evaluating the density on the grid we drop $ \hat g\_{a}^{(r)}(\mathbf{r}) $
>  and $ g^{(r)}\_{abo} (\mathbf{r}) $ and expand the density linearly as $\rho (\mathbf{r}) = \sum\_{a \in \mathcal{N}} \hat g^{(l)}\_{a}(\mathbf{r}) + \sum\_{(a, b) \in \mathcal{E}\_\text{e}} \sum^{N^o}\_{o} g^{(l)}\_{abo}(\mathbf{r}) = \sum\_{a \in \mathcal{N}} \sum\_{\mu} \hat g^{(l)}\_{a\mu} \omega^{Z\_a}\_\mu(\mathbf{r} - \mathbf{r}_a) + \sum\_{(a, b) \in \mathcal{E}\_\text{e}} \sum^{N^o}\_{o} \sum\_{\mu} g^{(l)}\_{abo\mu} \omega^{Z\_a}\_\mu(\mathbf{r} - \mathbf{r}_a)$ where the sum over $b$ can be carried out before evaluating the basis functions $\omega^{Z\_a}\_\mu(\mathbf{r} - \mathbf{r}_a)$ on the grid. Three models were trained on the QM9 VASP data using this linear expansion and evaluated on the test set. The mean and standard error are reported:
>
> |  | NMAE [%] |
> |:-----------|:-------------------:|
> | Linear Expansion  | 0.2716 ± 0.0007 |
> | Quadratic Expansion | 0.1381 ± 0.0003 |
>
> The quadratic expansion clearly outperforms the linear expansion.
>
> ### Basis Overlap Attention
>
> To evaluate the impact of choosing BOA instead of some other architecture as backbone, we train a BOA model using the same basis set as employed in one of the settings reported in [1] for SCDP. To isolate the impact of the architecture choice, we employ the linear expansion also used for the previous ablation. BOA achieves better performance than the corresponding SCDP model:
>
> |  | NMAE [%] |
> |:-----------|:-------------------:|
> | SCDP  | 0.504 |
> | BOA linear even-tempered basis | 0.393 |
>
>
> ### Learned Radial Correction
> An experiment studying the impact of the learned radial correction could already be found in Appendix Section A.1.1 in the original submission. We added a reference to this ablation in Section 2.9.

---

> > ### Author Response · Authors · 2025-11-22
> > **Answer to Reviewer igug (continued)**
> >
> > ### Efficiency
> > >(Efficiency) Have you evaluated the computational efficiency of BOA in terms of speed and memory usage compared to ELECTRA or other recent electron-density models?
> >
> > We evaluate the inference efficiency of ELECTRA [2], SCDP [1], BOA, and BOA with *def2-TZVP* basis models. We note that there is no difference in inference efficiency between the small and large
> > settings, since only the number of probe points and the batch size during training are changed, and we therefore do not
> > need to differentiate between them here. All models where evaluated on a 40GB A100 GPU using a block size maximizing the VRAM usage. The time per molecule is evaluated over the whole QM9 VASP test set and the mean and standard deviation over these molecules are reported:
> >
> > | Model | Time per molecule [s] | NMAE |
> > |:-----------|:-------------------:|:-------------------:|
> > | ELECTRA  | 0.14 ± 0.03 | 0.177 |
> > | SCDP       | 0.58 ± 0.16 | 0.178 |
> > | BOA    | 1.27 ± 0.27 | 0.1339 ± 0.0005 |
> > | BOA *def2-TZVP* | 0.64 ± 0.15 | 0.1504 ± 0.0003 |
> >
> > While BOA is significantly slower than ELECTRA and SCDP, using the smaller *def2-TZVP* mitigates this effect to some extent and achieves a lower error than the previous state of the art model with efficiency comparable to SCDP. The efficiency/accuracy tradeoff is easily steerable in BOA by employing basis sets of different sizes.
> >
> > > The large model reportedly requires ∼94 GB GPU memory, which raises questions about scalability and computational efficiency.
> >
> > It is true that memory requirements for the *BOA large* setting are high, with limited gain in terms of accuracy. Our practical recommendation would be to use *BOA small*, which can easily be trained on a 40GB GPU, making the memory requirements less restrictive. Additionally, as reported in the *Basis set choice* section, the *def2-TZVP* basis set also gives very competitive results, but with a much smaller number of basis functions. Such a model can be trained on a 20GB GPU, with only slightly worse results. Using these different settings, the memory requirements are easily steerable and can when necessary be greatly reduced. Additionally, during inference memory requirements are not problematic, since the density can be evaluated in a block-wise manner on the grid, making it easy to reduce the memory requirements to what is available.
> >
> > ### Free index in equations
> > > Eq. (13) and others: Consider to change $(a,b) \in \mathcal{E}$ to $b \in \mathcal{N}(a)$ or equivalent so that $a$ remains a free index.
> >
> > We agree and have changed the submission accordingly.
> >
> >
> > ### Non-atom-centric extensions
> > >(Non–atom-centric extensions) Could BOA be extended to incorporate non–atom-centered or floating basis functions, similar to ELECTRA’s floating orbitals, to further enhance flexibility?
> >
> > This is a very interesting suggestion; we indeed think that this could be an interesting avenue for future work. Incorporating non-atom-centered basis functions with fixed positions, e.g. on the bonds, like what is employed by SCDP [1] is easy from a technical standpoint. This would of course lead to an even larger number of basis functions per molecule, which could pose efficiency challenges. A significantly larger challenge would be the incorporation of floating basis functions with predicted positions. Since BOA uses the overlap matrix of the basis functions in the forward pass, a fully differentiable implementation of the calculation of the overlap matrix would be needed to capture the full impact of the changes of basis function positions. While technically challenging, this could be possible, e.g. by employing the differentiable package PySCFAD [3] which enables fully differentiable calculation of overlap integrals using Jax [4].
> >
> > ### Relation to KS-DFT coefficients
> > > (Relation to KS-DFT coefficients) Do the predicted coefficients correspond directly to the Kohn–Sham orbital coefficient matrix, or are they only related to the density-matrix blocks derived from it?
> >
> > While the quadratic expansion of the density is inspired by the Kohn-Sham coefficient matrix, the predicted coefficients will not directly correspond to the Kohn-Sham coefficient matrix for multiple reasons. When the smooth absolute value or the radial correction is employed, there is no direct correspondence between the basis functions employed in the Kohn-Sham calculations and the basis functions employed in BOA anymore, even if the same basis set is used originally. But more importantly, even when smooth absolute value and radial correction are not used there is no direct correspondence. The key point here is that the Kohn-Sham density matrix must fulfill additional constraints that are not enforced in the coefficients predicted by BOA, i.e. the density matrix is low rank and idempotent, using the overlap matrix as metric, owing to how it is built from the molecular orbitals. Not enforcing these constraints adds expressivity and is unproblematic when only the electron density is predicted.

---

> ### Author Response · Authors · 2025-11-22
> **Answer to Reviewer igug (continued)**
>
> ### Scalability
> >(Scalability) What is the computational scaling behavior of BOA with respect to the number of atoms or basis functions? Have you tested the model’s scalability on larger molecular systems?
>
> Due to the cutoff employed in BOA, the forward pass scales linearly with the system size, at least when also the overlap matrix is calculated in a block-wise manner. Here, we calculate the full overlap matrix since this has no large impact on computational efficiency; a block-wise calculation is unproblematic. We performed additional experiments on molecules taken from the QMugs [5] dataset, the results of which are shown in Figure 4 in the revised submission. We use two models trained on QM9 PySCF data [6] for this evaluation. One model is trained using the standard *BOA large* settings while a second model is trained employing smaller cutoffs. The message passing cutoff was lowered to $r_{mp} = 3$Å and the edge feature cutoff was lowered to $r_{e} = 2$Å for this model. This reduction is necessary to make generalization possible, as the distribution shift using a large field of view seems too large when generalizing from small to large molecules. This can clearly be seen in the results in Figure 4 in the revised manuscript, showing the time needed, and the accuracy achieved by the two BOA models and the ResNet [6] when applied to larger molecules of up to approximately 200 atoms. While the BOA model with standard cutoff and the ResNet have similar inference times, the standard BOA model performs worse on larger molecules with normalized mean absolute density errors of nearly $2$%. The BOA model with smaller cutoff however clearly outperforms both the standard BOA and the ResNet models, showing roughly constant normalized mean absolute errors of less than $0.5$% for all molecules, demonstrating remarkable generalization capabilities while having lower inference times than the other two models.
>
> We appreciate the helpful comments and hope that we have addressed your concerns. We are looking forward to any further discussion.
>
> [1] Fu, Xiang, Andrew Scott Rosen, Kyle Bystrom, et al. “A Recipe for Charge Density Prediction.” Paper presented at The Thirty-eighth Annual Conference on Neural Information Processing Systems. November 6, 2024. https://openreview.net/forum?id=b7REKaNUTv.
>
> [2] Elsborg, Jonas, Luca Thiede, Alán Aspuru-Guzik, Tejs Vegge, and Arghya Bhowmik. “ELECTRA: A Cartesian Network for 3D Charge Density Prediction with Floating Orbitals.” arXiv:2503.08305. Preprint, arXiv, May 19, 2025. https://doi.org/10.48550/arXiv.2503.08305.
>
> [3] Zhang, Xing, and Garnet Kin-Lic Chan. “Differentiable Quantum Chemistry with PySCF for Molecules and Materials at the Mean-Field Level and Beyond.” The Journal of Chemical Physics 157, no. 20 (2022): 204801. https://doi.org/10.1063/5.0118200.
>
> [4] Bradbury, James, Roy Frostig, Peter Hawkins, et al. JAX: Composable Transformations of Python+NumPy Programs. V. 0.3.13. Released 2018. http://github.com/jax-ml/jax.
>
> [5] Isert, Clemens, Kenneth Atz, José Jiménez-Luna, and Gisbert Schneider. “QMugs, Quantum Mechanical Properties of Drug-like Molecules.” Scientific Data 9, no. 1 (2022): 273. https://doi.org/10.1038/s41597-022-01390-7.
>
> [6] Li, Chenghan, Or Sharir, Shunyue Yuan, and Garnet Kin-Lic Chan. “Image Super-Resolution Inspired Electron Density Prediction.” Nature Communications 16, no. 1 (2025): 4811. https://doi.org/10.1038/s41467-025-60095-8.

---

### Author Response · Authors · 2025-11-23
**General Answer to all Reviewers**

We thank all reviewers for their helpful comments and questions. Here we want to address two small changes made in the revised submission.

Since the original submission, upon request, the kind authors of [1] agreed to share the split used for their evaluation on the QM9 PySCF data. We reran the experiments on this data using this new split, making the results more comparable to the ResNet model.

Additionally, we were able to run the BOA large model five times on both QM9 PySCF and QM9 VASP and we added the statistics for the results of these models to the submission.

[1] Li, Chenghan, Or Sharir, Shunyue Yuan, and Garnet Kin-Lic Chan. “Image Super-Resolution Inspired Electron Density Prediction.” Nature Communications 16, no. 1 (2025): 4811. https://doi.org/10.1038/s41467-025-60095-8.

---

### Author Response · Authors · 2025-12-03
**Revision Summary**

We again thank all Reviewers for their helpful comments and questions. We have addressed all of these and we summarize the salient changes and findings here: We

1. evaluated BOA on the Coulomb energy of the predicted densities and the reduction in SCF iterations that is achieved when starting Kohn-Sham DFT calculations with the predicted densities instead of a standard initial guess, demonstrating that the state-of-the-art density error achieved by BOA readily translates to physical properties derived from the predicted densities (App. A.2).

2. performed an additional experiment evaluating the impact of the choice of basis set, showing that BOA achieves competitive results even with smaller basis sets (App. A.1.2).

3. conducted an ablation experiment to ensure the benefit of using a quadratic expansion of the density instead of a linear expansion (App. A.1.3).

4. evaluated the efficiency of BOA in comparison to previous state-of-the-art models (App. A.3).

5. applied BOA to large molecules of up to 200 atoms, showing scalability to large structures while maintaining small errors in the predicted ground state electron density even when only training on small QM9 molecules (Sec. 3.2).

6. performed an ablation experiment to gauge the impact of the chosen cutoff, demonstrating that a larger cutoff only slightly improves the electron density prediction (App. A.1.4).

7. discussed the spatial distribution of the error in the predicted density (App. A.4).

8. introduced an additional figure, illustrating the basis overlap message passing procedure introduced in this work (Figure 2 in the revised submission).


We appreciate all your input and regret that further discussion was not possible due to the unfortunate circumstances. We truly hope we were able to satisfy any concerns.

---

### Meta-Review · Area_Chair_v4QW · 2026-01-11

**Summary:**

The authors present an equivariant graph neural network for predicting electron densities, bypassing the expensive iterative method used in Kohn-Sham DFT calculations. The main idea is to use the outer product of basis set coefficients for orbitals as input features. Despite the quadratic cost of computing the overlap matrix, the authors show that the method can be extended to reasonably large systems. The method is shown to be more accurate on the QM9 dataset than existing methods. I recommend acceptance.

**Reviewer Concerns:**

The reviewers were concerned that the method was only used for predicting charge density and not other observables like energy. Follow-up experiments by the authors during review addressed these concerns.

**Reviewer Scores:**

It's possible that some of the reviewers might have raised their scores.

---

### Decision · Program_Chairs · 2026-01-26

Accept (Poster)